# Identification of FAM53C as a cytosolic-anchoring inhibitory binding protein of the kinase DYRK1A

Yoshihiko Miyata ⓘ, Eisuke Nishida

The protein kinase DYRK1A encoded in human chromosome 21 is the major contributor to the multiple symptoms observed in Down syndrome patients. In addition, DYRK1A malfunction is associated with various other neurodevelopmental disorders such as autism spectrum disorder. Here, we identified FAM53C with no hitherto known biological function as a novel suppressive binding partner of DYRK1A. FAM53C is bound to the catalytic protein kinase domain of DYRK1A, whereas DCAF7/WDR68, the major DYRK1A-binding protein, binds to the N-terminal domain of DYRK1A. The binding of FAM53C inhibited autophosphorylation activity of DYRK1A and its kinase activity to an exogenous substrate, MAPT/Tau. FAM53C did not bind directly to DCAF7/WDR68, whereas DYRK1A tethered FAM53C and DCAF7/WDR68 by binding concurrently to both of them, forming a tri-protein complex. DYRK1A possesses an NLS and accumulates in the nucleus when overexpressed in cells. Co-expression of FAM53C induced cytoplasmic re-localization of DYRK1A, revealing the cytoplasmic anchoring function of FAM53C to DYRK1A. Moreover, the binding of FAM53C to DYRK1A suppressed the DYRK1A-dependent nuclear localization of DCAF7/WDR68. All the results show that FAM53C binds to DYRK1A, suppresses its kinase activity, and anchors it in the cytoplasm. In addition, FAM53C is bound to the DYRK1A-related kinase DYRK1B with an Hsp90/Cdc37-independent manner. The results explain for the first time why endogenous DYRK1A is distributed in the cytoplasm in normal brain tissue. FAM53C-dependent regulation of the kinase activity and intracellular localization of DYRK1A may play a significant role in gene expression regulation caused by normal and aberrant levels of DYRK1A.

## Introduction

DYRK1A (dual-specificity tyrosine-phosphorylation regulated kinase 1A) is a proline-directed serine/threonine protein kinase belonging to the CMGC family (Becker & Joost, 1999). The amino acid sequence of the catalytic domain of DYRK1A is distantly related to that of MAP kinases (Miyata & Nishida, 1999), implying that DYRK1A may play roles in certain cellular signal transduction systems. DYRK1A phosphorylates various substrates both in the nucleus and cytoplasm and consequently acts as a regulator of the cell cycle, cell quiescence, and cell differentiation (Aranda et al, 2011; Becker & Sippl, 2011). DYRK1A is also involved in many other cellular processes such as cytoskeletal organization (Ryoo et al, 2007; Ori-McKenney et al, 2016) and DNA damage response (Guard et al, 2019; Menon et al, 2019; Roewenstrunk et al, 2019). Human DYRK1A is encoded in the Down Syndrome Critical Region in chromosome 21 (Galceran et al, 2003; Hämmerle et al, 2003), and higher expression of DYRK1A is responsible for most of the phenotypes including intellectual disability of Down syndrome patients (Altafaj et al, 2001). A role of DYRK1A has also been suggested in other pathological conditions observed in Down syndrome patients, such as earlier onset of Alzheimer disease (Kimura et al, 2007; Branca et al, 2017), type 2 diabetes (Shen et al, 2015; Wang et al, 2015), and craniofacial malformation (Blazek et al, 2015; McElyea et al, 2016; Redhead et al, 2023). In addition, recent studies suggest that DYRK1A is also involved in several other neurodevelopmental disorders including Attention Deficit Hyperactivity Disorder (Tian et al, 2019), Autism Spectrum Disorder (O'Roak et al, 2012; De Rubeis et al, 2014; van Bon et al, 2016), and DYRK1A-haploinsufficiency syndrome (Courcet et al, 2012; Duchon & Hérault, 2016; Courraud et al, 2021). Altogether, it is evident that DYRK1A plays a fundamental role in the process of neurodevelopment and neurofunction (Arbones et al, 2019; Atas-Ozcan et al, 2021). DYRK1A has thus recently emerged in the drug discovery field as an attractive therapeutic target kinase. In contrast to many signaling kinases whose activities are regulated by phosphorylation in the activation loop by upstream kinase-kinases, DYRK1A phosphorylates itself in an activation loop tyrosine residue (Tyr321) during the transitional translation process, and this autophosphorylation is essential for the constitutive serine/threonine-specific kinase activity of mature DYRK1A for exogenous substrates (Himpel et al, 2001; Lochhead et al, 2005). The precise molecular mechanism of DYRK1A regulating cellular physiology and human disease conditions remains largely unknown.

Department of Cell and Developmental Biology, Graduate School of Biostudies, Kyoto University, Kyoto, Japan

Correspondence: ymiyata@lif.kyoto-u.ac.jp
Eisuke Nishida's present address is RIKEN Center for Biosystems Dynamics Research, Kobe, Japan

Intracellular distribution of DYRK1A is of critical importance and has been a matter of considerable debates. DYRK1A possesses NLS, and thus DYRK1A accumulates inside the nucleus when overexpressed in various cell lines (Becker et al, 1998; Álvarez et al, 2003; Miyata & Nishida, 2011). Many transcription factors, including NFAT, FOXO1, and STAT3, are controlled by DYRK1A-dependent phosphorylation in the nucleus (Arron et al, 2006; Gwack et al, 2006; Bhansali et al, 2021). DYRK1A interacts with RNA polymerase II in the nucleus and promotes its hyperphosphorylation in the C-terminal domain repeats through a phase separation mechanism (Di Vona et al, 2015; Lu et al, 2018; Yu et al, 2019). In addition, DYRK1A directly binds to chromatin regulatory regions to control gene expression (Di Vona et al, 2015; Li et al, 2018; Yu et al, 2019). These previous reports indicate that DYRK1A functions in the cell nucleus. On the other hand, endogenous DYRK1A has been often observed in the cytoplasmic and cytoskeletal fractions of cultured cells and natural brains of human and experimental animals (Martí et al, 2003; Wegiel et al, 2004; Ferrer et al, 2005; Aranda et al, 2008; Nguyen et al, 2018). These observations suggest that DYRK1A should also have cytoplasmic substrates. DYRK1A plays a critical role in activating the mitochondrial import machinery by cytoplasmic phosphorylation of the import receptor TOM70 (Walter et al, 2021), indicating that cytoplasmic function of DYRK1A is also physiologically important. Taking the constitutive property of DYRK1A activity into account, the intracellular distribution of DYRK1A should be strictly regulated by an unknown molecular mechanism.

We and others previously identified DCAF7 (DDB1 and CUL4 Associated Factor 7) (also called as WDR68 [WD Repeat protein 68] or HAN11 [Human homolog of ANthocyanin regulatory gene 11], and we use "DCAF7/WDR68" throughout the text hereafter) as a well-conserved major binding partner for DYRK1A (Skurat & Dietrich, 2004; Morita et al, 2006; Mazmanian et al, 2010; Miyata & Nishida, 2011). Structural analysis indicates that DCAF7/WDR68 forms a seven-propeller ring structure (Miyata et al, 2014), suggesting that DCAF7/WDR68 plays a role in tethering numerous binding partners to its structure as do other WD40-repeat proteins. In fact, several proteins, including IRS1, E1A oncoprotein, and RNA polymerase II, have been shown to associate with DYRK1A via DCAF7/WDR68 (Glenewinkel et al, 2016; Yu et al, 2019; Frendo-Cumbo et al, 2022). Likewise, certain proteins may make complexes with DCAF7/WDR68 via DYRK1A. Our earlier phospho-proteomic study obtained more than 250 associating partner candidates for DCAF7/WDR68 (Miyata et al, 2014), which may include uncharacterized DYRK1A-binding proteins.

DYRK1B is the closest relative of DYRK1A with 85% amino acid identities in the catalytic protein kinase domain (Becker et al, 1998), and most of DYRK1A-interacting partners including DCAF7/WDR68 are shared with DYRK1B (Miyata & Nishida, 2011; Varjosalo et al, 2013). On the other hand, Hsp90 (Heat Shock Protein 90) and Cdc37 (Cell Division Cycle Protein 37) make a stable complex only with DYRK1B but not with DYRK1A (Miyata & Nishida, 2021). Identification of additional interacting partners for DYRK1A and DYRK1B is of critical importance to understand the physiological roles of these kinases.

In this study, we have identified FAM53C as a specific binding partner for DYRK1A and DYRK1B. The catalytic kinase domain of DYRK1A was responsible for the FAM53C binding, while DCAF7/

WDR68 binds to the N-terminal domain of DYRK1A as previously shown (Miyata & Nishida, 2011; Glenewinkel et al, 2016). Hence, DYRK1A could simultaneously bind to both DCAF7/WDR68 and FAM53C, forming a tri-protein complex, demonstrating a tethering function of DYRK1A. FAM53C binding suppressed the protein kinase activity of DYRK1A. In addition, the binding of FAM53C induced cytoplasmic retention of DYRK1A, and the balance between the levels of DYRK1A and FAM53C determined the intracellular distribution of DYRK1A. These results indicate that FAM53C anchors DYRK1A in the cell cytoplasm in an inactive state. FAM53C may be a key molecule which resolves the long-lasting controversial discrepancy of the intracellular distribution of DYRK1A between overexpressed cell lines and the endogenous setting in human brain.

# Results

## Identification of FAM53C as a DCAF7/WDR68–DYRK1A interactor

DCAF7/WDR68 is a major binding partner of DYRK1A and DYRK1B (Skurat & Dietrich, 2004; Morita et al, 2006; Mazmanian et al, 2010; Ritterhoff et al, 2010; Miyata & Nishida, 2011; Glenewinkel et al, 2016). By a phospho-proteome analysis, we have previously identified proteins that are physically associated with DCAF7/WDR68, directly or indirectly, along with their phosphorylation sites (Miyata et al, 2014). The most prominent binding partner for DCAF7/WDR68 is a molecular chaperone complex TriC/CCT (Miyata et al, 2014), and some of identified DCAF7/WDR68-associated partners, such as actin and tubulin, may therefore interact with DCAF7/WDR68 indirectly via TRiC/CCT. Similarly, DYRK1A-binding proteins should possibly be included in the DCAF7/WDR68 interactome network. We identified FAM53C, FAMily with sequence similarity 53 member C, in the DCAF7/WDR68-associated phosphoprotein list. The mass analysis identified nine peptides with one phosphoserine each, and they covered 32.1% (126aa in 392aa) of FAM53C in total (Fig 1A). This result indicates that FAM53C is an interacting partner for DCAF7/WDR68.

We then examined the BioPlex (biophysical interactions of ORFeome-based complexes) protein-protein interaction database generated by immunoprecipitation of proteins stably expressed in human cell lines followed by mass spectrometry (Huttlin et al, 2017, 2021). FAM53C (orange) is included in the protein-protein interaction network of DYRK1A (Fig 1B, upper panels), DCAF7/WDR68 (Fig 1B, middle panels), and DYRK1B (Fig 1B, lower right panel). In addition, DYRK1A (blue), DYRK1B (magenta), and DCAF7/WDR68 (yellow) are all included in the FAM53C-interacting network (Fig 1B, lower left panel). These protein-protein network analyses convincingly strengthen our result that FAM53C is associated with DCAF7/WDR68 and also with DYRK1A and DYRK1B.

Human FAM53C consists of 392 amino acids and is rich in Ser (16%), Pro (14%), Leu (10%), and Arg (9%); thus, half of the protein consists of only the four amino acids, making FAM53C a low-complexity sequence protein. Orthologous proteins of FAM53C are encoded in all the mammals examined with high amino acid identities (99% in chimpanzee down to 80% in opossum and

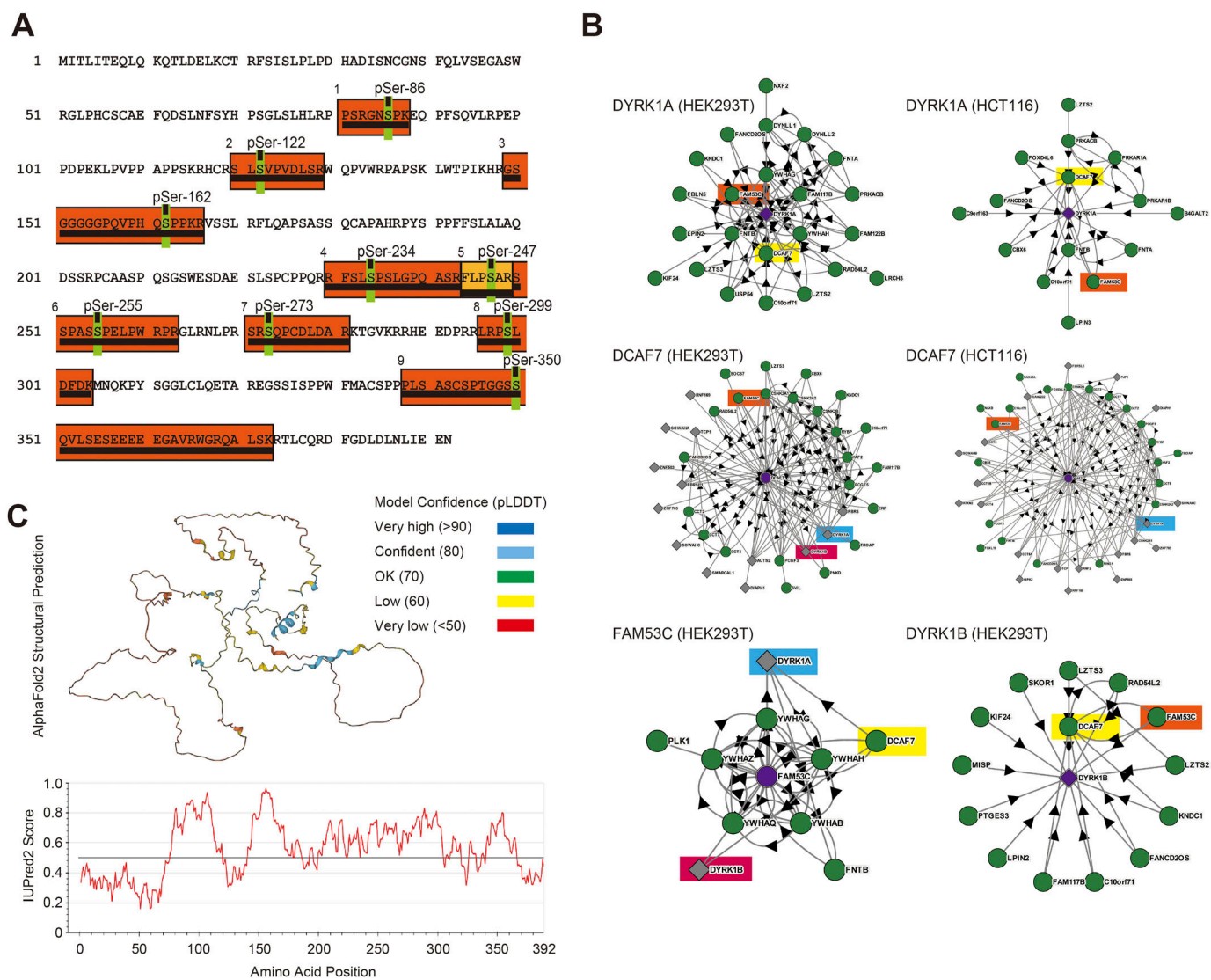

**Figure 1. Identification of FAM53C as an interactor candidate for DCAF7/WDR68 and DYRK1A.**
**(A)** Phospho-proteomic analysis of binding proteins of DCAF7/WDR68 exogenously expressed in COS7 cells identified nine phosphopeptides (underlined orange boxes) corresponding to FAM53C. The amino acid sequence of human FAM53C is shown with identified phosphorylation sites (shown in green with amino acid numbers). **(B)** BioPlex protein-protein interaction network analysis. The database was analyzed with DYRK1A (*upper panels*, in HEK293T [*left*] or HCT116 [*right*]), DCAF7/WDR68 (*middle panels*, in HEK293T [*left*] or HCT116 [*right*]), FAM53C (*lower left panel* in HEK293T), and DYRK1B (*lower right panel* in HEK293T) as queries (located in the center of the networkgrams). Identified interacting partners are shown and FAM53C (*orange*), DCAF7/WDR68 (*yellow*), DYRK1A (*blue*), and DYRK1B (*magenta*) are color-highlighted. Proteins with circles indicate baits, and proteins with diamonds indicate prey. The arrow heads show the bait-to-prey direction. **(C)** Structural characterization of FAM53C. Structural prediction of FAM53C by AlphaFold2 (*upper panel*) and the intrinsically disordered tendency score of FAM53C by IUPred2 (*lower panel*) are shown.

59% in platypus), with lower amino acid identities in lizard/snake (55%), frog (51–52%), turtle/crocodile (43–50%), shark/ray (38%), bonny fish (33–35%), and chicken (33%), but not in cnidarians, nematode, insects, yeast, or plants, suggesting that FAM53C may have vertebrate-specific functions. Query for the Phyre2 structural analysis (http://www.sbg.bio.ic.ac.uk/phyre2/index.cgi) of FAM53C resulted in no obvious similarity with known protein folds. A predicted structure of FAM53C in AlphaFold Protein Structure Database (https://alphafold.com/) shows that the most of FAM53C structure is highly flexible and thus not rigidly predictable. Prediction by the AlphaFold2 (alphafold2.ipynb: template mode = pdb70, unpaired + paired) indicates that structures of only a limited

number of small fragments of FAM53C can be determined with high confidence (Fig 1C). In addition, secondary structure predictions with several algorithms and intrinsically unstructured scores by IUPred2 (https://iupred2a.elte.hu/) both suggest that this protein is composed of coil structures with intrinsically disordered regions throughout the molecule, except for short helices in the N- and C-terminal edges and several possible tiny beta-strands (Fig 1C). According to these analyses, FAM53C seems to be a substantially disordered protein with a highly flexible structure. Two related proteins, FAM53A and FAM53B, are encoded in the human genome and the amino acid identities of FAM53C with FAM53A and FAM53B are 35% and 30%, respectively. FAM53C, as well as FAM53A and

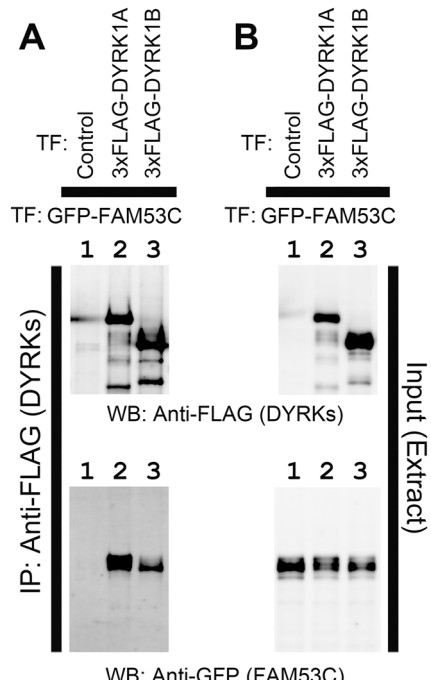

**Figure 2. Association of FAM53C with DYRK1A and DYRK1B.**
DYRK1A and DYRK1B were expressed as 3xFLAG-tagged proteins with GFP-tagged FAM53C in COS7 cells. DYRK1A and DYRK1B were immunoprecipitated with resin conjugated with anti-FLAG antibody and protein complexes were analyzed by SDS–PAGE/Western blotting. **(A)** The amounts of DYRK1A, DYRK1B (*upper panel*), and FAM53C (*lower panel*) in the immunocomplexes were shown by Western blotting. *Lane 1*, Control; *lane 2*, DYRK1A; *lane 3*, DYRK1B. **(B)** The amounts of DYRK1A, DYRK1B (*upper panel*), and FAM53C (*lower panel*) in the extracts were shown by Western blotting. *Lane 1*, Control; *lane 2*, DYRK1A; *lane 3*, DYRK1B. FAM53C were transfected in *lanes 1–3*.

FAM53B, have only very limited biochemical and physiological annotations for their function so far.

The human protein atlas database (https://www.proteinatlas.org/ENSG00000120709-FAM53C) indicates that FAM53C is expressed richly in the brain and bone marrow but is also expressed in many human tissues with low tissue specificity. The International Mouse Phenotyping Consortium database indicates that *FAM53C* knock-out mice are viable, showing a phenotype classified as "Decreased exploration in new environment" (https://www.mousephenotype.org/data/genes/MGI:1913556), suggesting that FAM53C may play a role in neuro-development and/or neurological function.

### Binding of expressed and endogenous FAM53C with DYRK1A and DYRK1B

As described above, we identified FAM53C as a DCAF7/WDR68-interacting protein, and large-scale interactome databases suggest that FAM53C may bind to DYRK1A and DYRK1B. We thus examined if FAM53C makes complexes with DYRK1A and DYRK1B by co-immunoprecipitation experiments. DYRK1A and DYRK1B were expressed as 3xFLAG-tagged proteins in mammalian cultured COS7 cells (Fig 2B, *top*) with GFP-FAM53C (Fig 2B, *bottom*) and immunoprecipitated with anti-FLAG antibody (Fig 2A, *top*). The binding of GFP-FAM53C was examined by Western blotting with anti-GFP

antibody (Fig 2A, *bottom*). The results indicated that FAM53C bound to and was co-immunoprecipitated with both DYRK1A and DYRK1B.

To examine if endogenous FAM53C binds to DYRK1A and DYRK1B, we made an antibody against FAM53C by immunizing a rabbit with a KLH-conjugated peptide, CQQDFGDLDLNLIEEN, corresponding to amino acids 377–392 (the last 16 amino acids of the C-terminal end) of orangutan FAM53C. The sequence in this region is identical in FAM53C of almost all mammals from opossum to primates. Very few exceptions with one amino acid change are observed in this region of FAM53C of human (Q379R), hylobates (G383R), and echidna/platypus (C377S). Similar sequences are not contained within any other mammalian proteins including the related family proteins FAM53A and FAM53B. The antiserum was purified on an affinity column of resin conjugated with the antigen peptide. The obtained antibody (Fig 3A, *top right panel*), but not pre-immune serum (Fig 3A, *top left panel*), recognized both monkey endogenous (*lane 1*) and exogenously expressed human FAM53C tagged with 3xFLAG (*lane 2*) or with GFP (*lane 3*). COS7 cells were then transfected with 3xFLAG-DYRK1A (Fig 3B, *lanes 3 & 4*) or 3xFLAG-DYRK1B (Fig 3B, *lanes 5 & 6*), and the binding of endogenous FAM53C was examined by co-immunoprecipitation experiments. The binding of endogenous FAM53C was evident in DYRK1A (*lane 4*) and DYRK1B (*lane 6*) im-munoprecipitates, only when cells expressed DYRK1A or DYRK1B and immunoprecipitated with anti-FLAG antibody, but not with control antibody (Fig 3B, *lanes 1–3 & 5*). The antibody against FAM53C recognized several protein bands in the extracts (Fig 3A, *top right panel*, *lane 1*), and the uppermost band shown by an asterisk corresponds to the FAM53C bound to DYRK1A and DYRKB in Fig 3B, suggesting that this band is full-length endogenous FAM53C and a couple of other lower bands might be proteolytic fragments of FAM53C and/or other proteins non-specifically recognized by the antibody. Altogether, these results show that FAM53C binds to both DYRK1A and DYRK1B.

### FAM53C binds to the catalytic kinase domain of DYRK1A

The FAM53C-binding domain in DYRK1A was next determined by co-immunoprecipitation experiments. WT and deletion mutants of 3xFLAG-tagged DYRK1A were expressed (Fig 4D) with GFP-FAM53C (Fig 4C) and the binding of FAM53C (Fig 4A) to immunoprecipitated DYRK1A (Fig 4B) was determined by Western blotting with anti-GFP antibody. The kinase domain (aa156–479) alone (DYRK1A(K), *lane 3*), but not the N-terminal (aa1-158) (DYRK1A(N), *lane 2*) nor C-terminal (aa481–763) (DYRK1A(C), *lane 4*) domain of DYRK1A, bound to FAM53C. Deletion of the C-terminal domain (DYRK1A(N+K), *lane 5*) or the N-terminal domain (DYRK1A(K+C), *lane 6*) of DYRK1A did not abolish the FAM53C binding. Sufficient levels of GFP-FAM53C expression were observed in all the extracts used for the co-immunoprecipitation assays (Fig 4C). The expression levels (Fig 4D) and the amounts of immunoprecipitated proteins (Fig 4B) of DYRK1A fragments differed with each other; thus, it was difficult to accurately estimate the difference of binding levels of DYRK1A domains to FAM53C. These results indicate that the catalytic kinase domain of DYRK1A, but not the N-terminal or C-terminal domain, is responsible for the FAM53C binding. A possible contribution of other parts of DYRK1A outside of the catalytic kinase domain for the FAM53C binding, however, cannot be excluded.

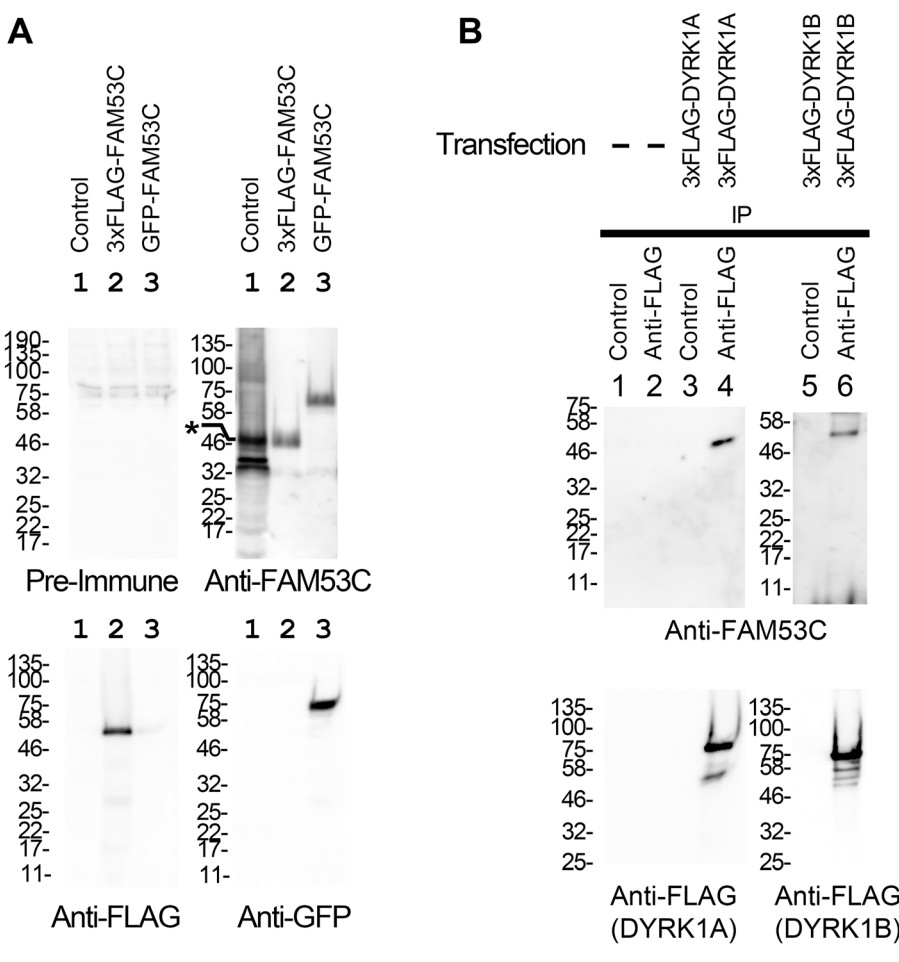

**Figure 3. Binding of endogenous FAM53C with DYRK1A and DYRK1B.**
**(A)** COS7 cells (*lane 1, control*) were transfected with 3xFLAG-FAM53C (*lane 2*) or GFP-FAM53C (*lane 3*). Total cell lysates were prepared and examined by Western blotting with indicated antibodies as follows. *Top left*, control pre-immune serum; *top right*, C-terminal peptide-directed FAM53C antibody (extracts in *lanes 2 & 3* in this panel were ×100 diluted to avoid signal saturation. The asterisk indicates the position of full length endogenous FAM53C); *bottom left*, anti-FLAG antibody, showing the expression of 3xFLAG-FAM53C; *bottom right*, anti-GFP antibody, showing the expression of GFP-FAM53C. **(B)** COS7 cells were transfected with 3xFLAG-tagged DYRK1A or DYRK1B and the co-immunoprecipitation of endogenous FAM53C with DYRK1A and DYRK1B was examined by Western blotting. *Lane 1*, control IgG-immunoprecipitate from non-transfected control cells; *lane 2*, anti-FLAG immunoprecipitate from non-transfected control cells; *lane 3*, control IgG-immunoprecipitate from 3xFLAG-DYRK1A-transfected cells; *lane 4*, anti-FLAG-immunoprecipitate from 3xFLAG-DYRK1A-transfected cells; *lane 5*, control IgG-immunoprecipitate from 3xFLAG-DYRK1B-transfected cells; *lane 6*, anti-FLAG-immunoprecipitate from 3xFLAG-DYRK1B-transfected cells. Western blotting images with the anti-FAM53C antibody (*upper panels*) and with anti-FLAG antibody (*lower panels*) are shown.

## Inhibition of the protein kinase activity of DYRK1A by the FAM53C binding

The effect of the FAM53C binding to the catalytic kinase domain of DYRK1A on its protein kinase activity was then examined. DYRK1A was expressed as a 3xFLAG-tagged protein in cultured COS7 cells with or without FAM53C co-expression and purified with anti-FLAG affinity resin followed by the elution with the 3xFLAG peptide. An equal amount of DYRK1A, alone or in complex with FAM53C, was then incubated in the presence of $Mg^{2+}$-ATP with purified recombinant MAPT/Tau protein, a well-established DYRK1A substrate (Woods et al, 2001). The levels of DYRK1A-dependent phosphorylation of MAPT/Tau on Thr212 were quantified by Western blotting with specific anti-pTau Thr212 antibody. As shown in Fig 5A, MAPT/Tau phosphorylation by DYRK1A was much lower when DYRK1A was in complex with FAM53C (*lane 6*) as compared to DYRK1A alone (*lane 5*). The total amount of MAPT/Tau protein was not altered by the incubation for kinase reactions, as shown by CBB staining (Fig 5B). The electrophoretic mobilities of DYRK1A were affected when DYRK1A was associated with FAM53C during the kinase reactions. In the absence of expressed FAM53C, a band with a slower mobility (upper band) of DYRK1A was detected (Fig 5C, *lane 5*), while this

upper band was not observed when DYRK1A was associated with overexpressed FAM53C (Fig 5C, *lane 6*). The slower mobility (the upper band) of DYRK1A in SDS–PAGE was previously ascribed to phosphorylated species of DYRK1A (Alvarez et al, 2007). Therefore, the slower mobility DYRK1A observed after the kinase reaction (*lane 5*) should be a result of DYRK1A autophosphorylation, and the absence of the phosphorylated DYRK1A band indicates that DYRK1A lost its autophosphorylation activity when bound to FAM53C. Specific binding of FAM53C to DYRK1A was detected by anti-FAM53C Western blotting only when both proteins were concurrently expressed and DYRK1A was isolated with specific affinity resin (Fig 5D, *lane 6*). The binding of endogenous FAM53C to DYRK1A, as shown in Fig 3B, was not visible at this exposure. MAPT/Tau phosphorylation and the DYRK1A signal were not detected in control conditions (Fig 5A and C, *lanes 1–4*). As shown in Fig 5E and G, the pTau (Thr212) signal and the DYRK1A mobility up-shift required both $Mg^{2+}$ and ATP (*lane 7–10*), validating that these are ascribed to the $Mg^{2+}$/ATP-dependent protein phosphorylation reaction. The amounts of MAPT/Tau protein stayed the same (Fig 5F). Taken together, these data indicate that DYRK1A possesses lower protein kinase activity toward both itself and an exogenous substrate when it is associated with FAM53C.

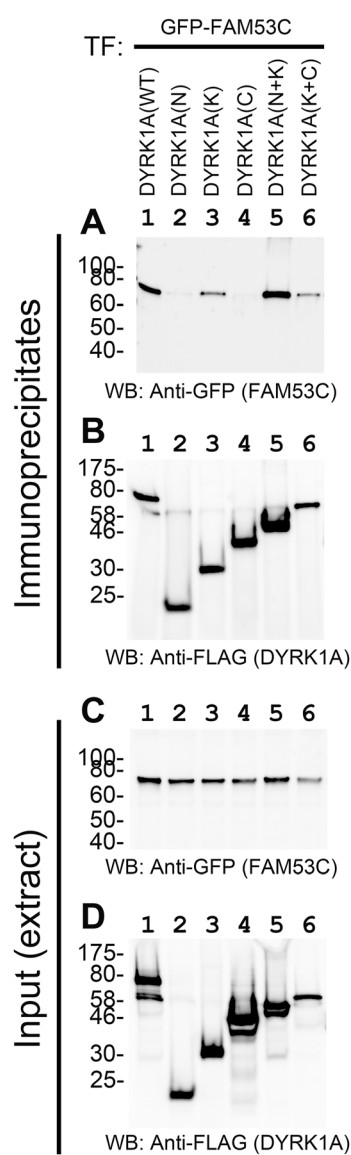

**Figure 4. Identification of the FAM53C-binding domain in DYRK1A.**
3xFLAG-DYRK1A (WT or deletion mutants) and GFP-FAM53C were expressed in COS7 cells and the binding of FAM53C to DYRK1A was examined by co-immunoprecipitation experiments. *Lane 1*, DYRK1A(WT); *lane 2*, DYRK1A(N); *lane 3*, DYRK1A(K); *lane 4*, DYRK1A(C); *lane 5*, DYRK1A(N+K); *lane 6*, DYRK1A(K+C). **(A)** Western blotting of immunoprecipitates with anti-GFP antibody for detection of DYRK1A-bound FAM53C. **(B)** Western blotting of immunoprecipitates with anti-FLAG antibody for detection of immunoprecipitated DYRK1A. **(C)** Expression levels of GFP-FAM53C in total cell lysates. **(D)** Expression levels of 3xFLAG-tagged WT and deletion mutants of DYRK1A in total cell lysates.

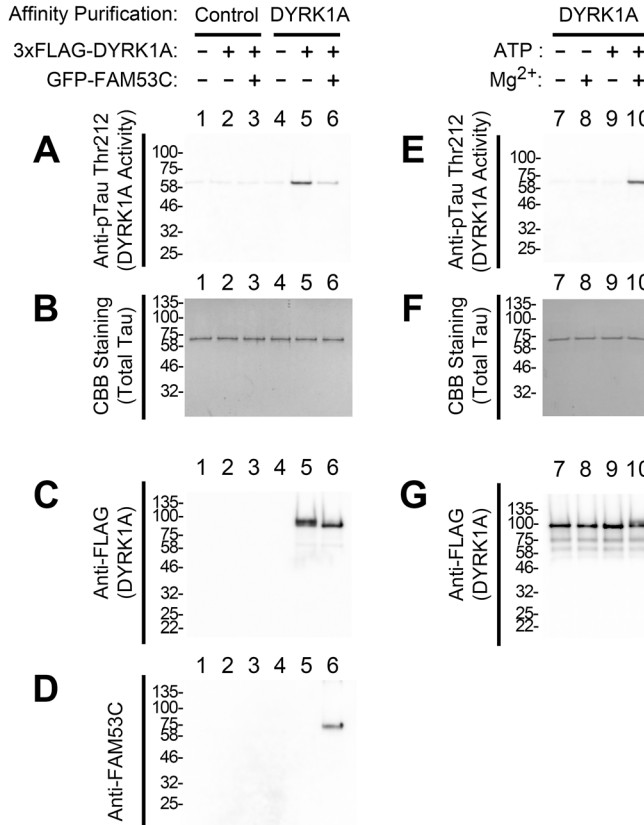

**Figure 5. Suppressive effect of the FAM53C binding on the kinase activity of DYRK1A.**
3xFLAG-DYRK1A was expressed in COS7 cells (lanes 2, 3, 5, & 6) with (lanes 3 & 6) or without (lanes 1, 2, 4, & 5) GFP-FAM53C and affinity purified. As controls, mock affinity purification with control resin (*lanes 1–3*) and affinity purification from non-transfected cell extracts (*lanes 1 & 4*) were included. In vitro DYRK1A protein kinase assay was conducted with recombinant MAPT/Tau protein as a substrate. In addition, $Mg^{2+}$/ATP requirements for the kinase reactions were examined (*lanes 7–10*). Purified recombinant MAPT/Tau was incubated with affinity-purified DYRK1A with or without $Mg^{2+}$ (10 mM) and/or ATP (5 mM) as indicated on the top. DYRK1A-dependent phosphorylation of MAPT/Tau on Thr212 and DYRK1A electrophoretic mobilities were determined by Western blotting. **(A)** Anti-phospho-Tau (Thr212) Western blotting showing the DYRK1A kinase activity to an exogenous substrate MAPT/Tau. **(B)** CBB staining of the kinase reaction mixtures showing the amounts of MAPT/Tau protein. **(C)** Anti-FLAG Western blotting showing the amounts and electrophoretic mobilities of DYRK1A. **(D)** Anti-FAM53C Western blotting showing the association of overexpressed FAM53C with DYRK1A. Endogenous FAM53C was not visible at this exposure. **(E)** Anti-phospho-Tau (Thr212) Western blotting showing the DYRK1A kinase activity to an exogenous substrate MAPT/Tau. **(F)** CBB staining of the kinase reaction mixtures showing the amounts of MAPT/Tau protein. **(G)** Anti-FLAG Western blotting showing the amounts and electrophoretic mobilities of DYRK1A.

## DYRK1A-dependent association of WDR68/DCAF7 with FAM53C

Proteomic as well as co-immunoprecipitation experiments indicate that FAM53C makes a complex with both DYRK1A and DCAF7/WDR68. We next set up an experiment to clarify if FAM53C directly binds to DYRK1A, DCAF7/WDR68, or both. DCAF7/WDR68 was expressed in COS7 cells as an HA-tagged protein with or without 3xFLAG-tagged FAM53C. In addition, GFP-tagged full-length or deletion mutants of DYRK1A were concurrently expressed. FAM53C and its associated proteins were immunoprecipitated, and then the binding of DCAF7/WDR68 and DYRK1A was examined by Western blotting. As in the case for the combination of 3xFLAG-DYRK1A and GFP-FAM53C (Figs 2 and 4), 3xFLAG-FAM53C and GFP-DYRK1A were found to be interacted (Fig 6B, *lane 4*). In addition, DCAF7/WDR68 was included in the FAM53C-DYRK1A complex (Fig 6C, *lane 4*). DCAF7/WDR68 could not be co-immunoprecipitated with FAM53C in the absence of DYRK1A (Fig 6C, *lane 3*), suggesting that FAM53C does not bind directly to DCAF7/WDR68. This result is in sharp contrast to the binding of

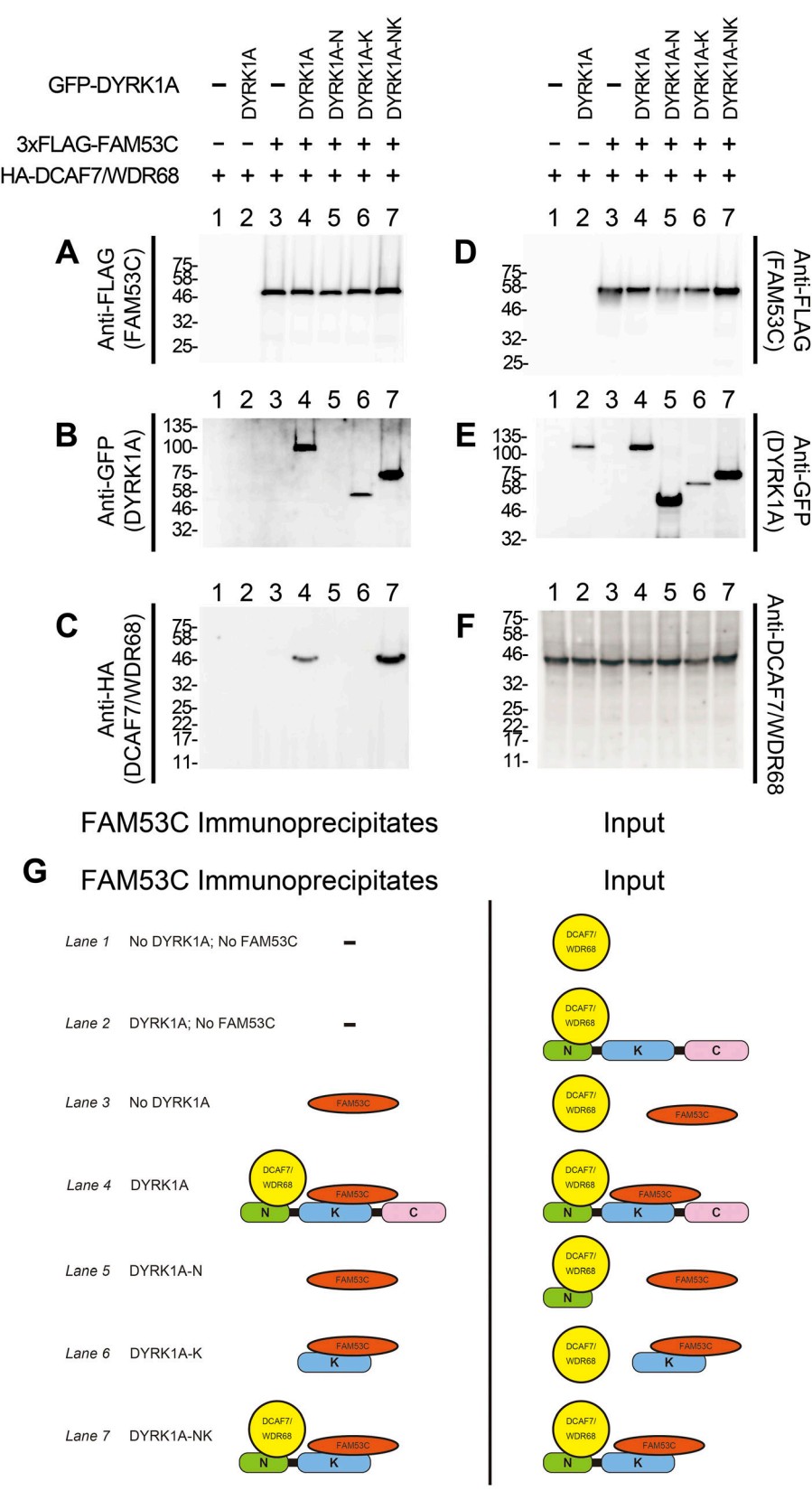

**Figure 6. DYRK1A-mediated association of FAM53C with DCAF7/WDR68.**
HA-DCAF7/WDR68 (*lanes 1–7*) and 3xFLAG-FAM53C (*lanes 3–7*) were expressed in COS7 cells with full length (*lanes 2 & 4*, DYRK1A), the N-terminal domain (*lane 5*, DYRK1A-N), the kinase domain (*lane 6*, DYRK1A-K), and N-terminal+kinase domain (*lane 7*, DYRK1A-NK) of GFP-tagged DYRK1A as indicated. **(A)** The amounts of immunoprecipitated FAM53C were shown by Western blotting with anti-FLAG antibody. **(B)** The association of DYRK1A with immunoprecipitated FAM53C was shown by Western blotting with anti-GFP antibody. **(C)** The association of DCAF7/WDR68 with immunoprecipitated FAM53C was shown by Western blotting with anti-HA antibody. **(D)** The amounts of total expressed FAM53C in cell extracts were shown by Western blotting with anti-FLAG antibody. **(E)** The amounts of WT and deletion mutants of expressed DYRK1A in cell extracts were shown by Western blotting with anti-GFP antibody. **(F)** The amounts of total expressed DCAF7/WDR68 in cell extracts were shown by Western blotting (anti-DCAF7/WDR68 antibody was used for this panel because of non-specific signals observed with anti-HA antibody in the cell extracts). **(G)** The schematic illustration of the tethering function of DYRK1A observed in (A, B, C). Expressed DCAF7/WDR68 (*yellow*), FAM53C (*red*), and DYRK1A (N-term [*green*], kinase [*blue*], and C-term [*pink*] domains) in the Input (total cell extracts) are shown (*right*). FAM53C, bound DYRK1A, and tethered DCAF7/WDR68 in the FAM53C immunocomplexes are indicated (*left*). **(A, B, C, D, E, F)** Lane numbers correspond to the lanes shown in (A, B, C, D, E, F).

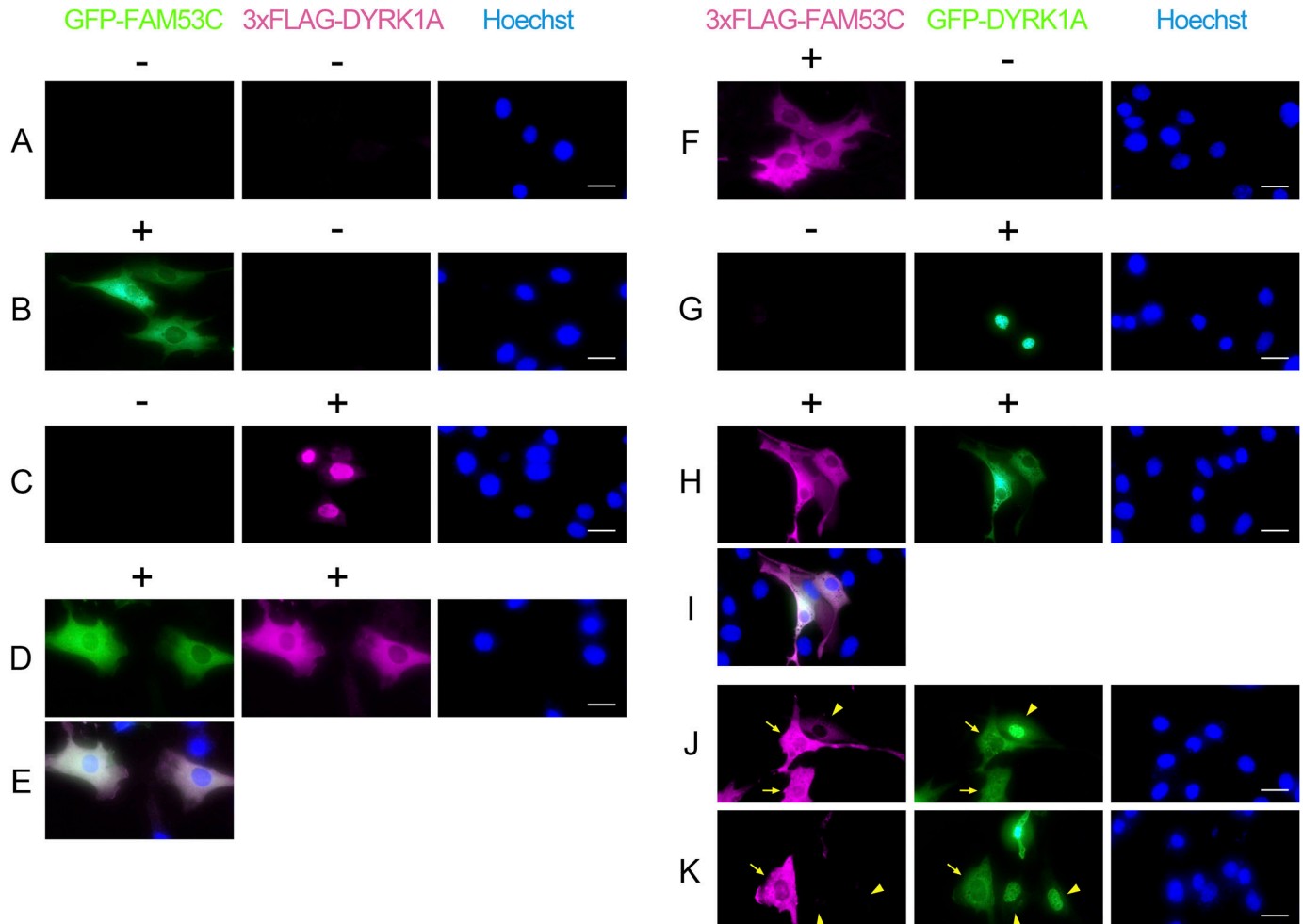

**Figure 7. FAM53C anchors DYRK1A in the cytoplasm.**
**(A, B, C, D, E)** NIH-3T3 cells were transfected with GFP-FAM53C and/or 3xFLAG-DYRK1A. The intracellular distribution of FAM53C (*left panels, green*) and DYRK1A (*center panels, magenta*) was visualized by fluorescent microscopy. Concurrently, cells were stained with Hoechst 33342 for visualization of nucleus (*right panels, blue*). **(A)** Control without transfection. **(B)** GFP-FAM53C alone. **(C)** 3xFLAG-DYRK1A alone. **(D)** Both FAM53C and DYRK1A. **(E)** A merged image of (D). Scale bars = 50 μm. **(F, G, H, I, J, K)** NIH-3T3 cells were transfected with 3xFLAG-FAM53C and/or GFP-DYRK1A. The intracellular distribution of FAM53C (*left panels, magenta*) and DYRK1A (*center panels, green*) was visualized by fluorescent microscopy. Concurrently, cells were stained with Hoechst 33342 for visualization of nucleus (*right panels, blue*). **(F)** 3xFLAG-FAM53C alone. **(G)** GFP-DYRK1A alone. **(H, I, J, K)** Both FAM53C and DYRK1A. **(I)** A merged image of (H). **(J, K)** Two representative microscopic fields showing cells with different expression levels of FAM53C. The arrows indicate cells with high FAM53C expression and arrowheads indicate cells with low or no FAM53C expression. Scale bars = 50 μm.

FAM53C to DYRK1A in the absence of exogenously expressed DCAF7/WDR68 (Figs 2–4). Additional co-expression of DYRK1A induced the association between FAM53C and DCAF7/WDR68 (Fig 6C, compare lanes 3 & 4), indicating that DYRK1A is required for the binding of DCAF7/WDR68 to FAM53C. Co-expression of DYRK1A-N domain, which binds to DCAF7/WDR68 but not to FAM53C (Fig 6B, lane 5), could not induce the association of DCAF7/WDR68 with FAM53C (Fig 6C, lane 5). Co-expression of DYRK1A-K domain, which does not bind to DCAF7/WDR68 but binds to FAM53C (Fig 6B, lane 6), could not induce the association of DCAF7/WDR68 with FAM53C (Fig 6C, lane 6). Contrarily, co-expression of DYRK1A-NK, which possesses both the DCAF7/WDR68-binding N-domain and the FAM53C-binding K-domain, induced association of DCAF7/WDR68 with FAM53C (Fig 6C, lane 7) along with DYRK1A-NK (Fig 6B, lane 7). Expression levels of FLAG-FAM53C, GFP-DYRKL1A, and HA-DCAF7/WDR68 are shown in Fig 6D–F. The amounts of immunoprecipitated FAM53C were shown in

Fig 6A. A schematic illustration of the expression of three proteins and FAM53C immunocomplexes is shown in Fig 6G. These results indicate that DCAF7/WDR68 is not able to bind directly to FAM53C without DYRK1A, and DCAF7/WDR68 binds to FAM53C in a DYRK1A-dependent manner. In other words, DYRK1A brings DCAF7/WDR68 and FAM53C together with its N-domain and K-domain, respectively.

## Intracellular distribution of FAM53C and DYRK1A

To elucidate the intracellular distribution of FAM53C, we expressed GFP-tagged FAM53C in mammalian cultured NIH-3T3 cells. The specific GFP signal indicated that FAM53C localized in cytoplasmic compartments of cells and was excluded from the nucleus (Fig 7B, green). 3xFLAG-DYRK1A when expressed alone localized in the nucleus (Fig 7C, magenta), as we and others have shown previously in several cell lines (Becker et al, 1998; Álvarez et al, 2003;

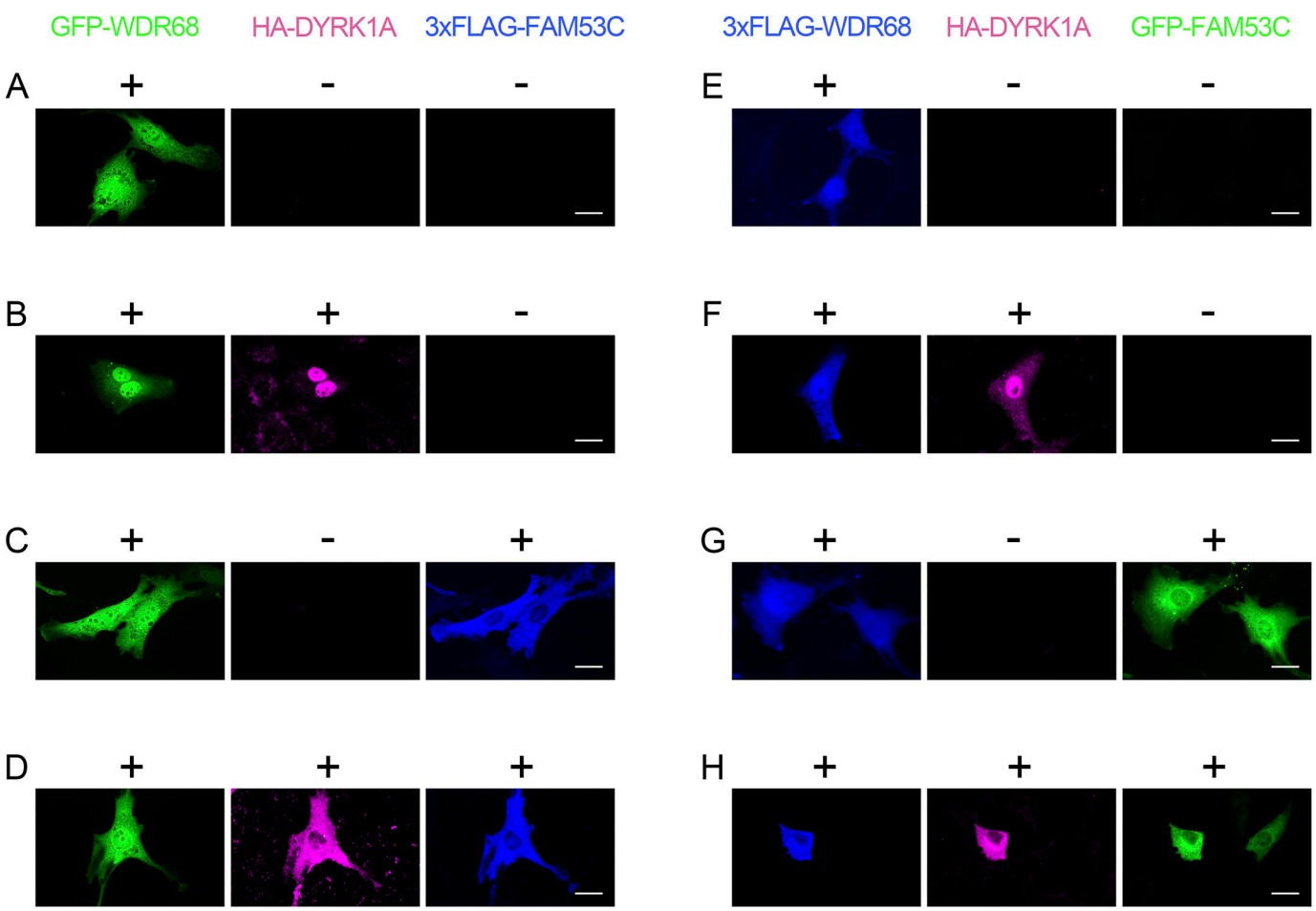

**Figure 8.  Tethering function of DYRK1A to FAM53C and DCAF7/WDR68.**
**(A, B, C, D)** NIH-3T3 cells were transfected with GFP-DCAF7/WDR68 (*green*), HA-DYRK1A (*magenta*), and 3xFLAG-FAM53C (*blue*) and the intracellular distribution of these proteins was examined by fluorescent microscopy. **(A)** DCAF7/WDR68 alone. **(B)** DCAF7/WDR68 and DYRK1A. **(C)** DCAF7/WDR68 and FAM53C. **(D)** DCAF7/WDR68, DYRK1A, and FAM53C. Scale bars = 50 *μm*. **(E, F, G, H)** NIH-3T3 cells were transfected with 3xFLAG-DCAF7/WDR68 (*blue*), HA-DYRK1A (*magenta*), and GFP-FAM53C (*green*), and the intracellular distribution of these proteins was examined by fluorescent microscopy. **(E)** DCAF7/WDR68 alone. **(F)** DCAF7/WDR68 and DYRK1A. **(G)** DCAF7/WDR68 and FAM53C. **(H)** DCAF7/WDR68, DYRK1A, and FAM53C. Scale bars = 50 *μm*.

Miyata & Nishida, 2011, 2021). Simultaneous expression of FAM53C with DYRK1A resulted in cytoplasmic re-distribution of DYRK1A (Fig 7D, *magenta*) and DYRK1A co-localization with FAM53C (Fig 7D and E), suggesting that FAM53C functions as a cytoplasmic anchoring protein for DYRK1A. The same conclusion was obtained when we exchanged the tags for FAM53C and DYRK1A as follows: 3xFLAG-FAM53C localized in the cytoplasm and was excluded from the nucleus (Fig 7F, *magenta*), while GFP-DYRK1A accumulated in the nucleus when expressed alone (Fig 7G, *green*). Co-expression of FAM53C with DYRK1A induced cytoplasmic retention of DYRK1A (Fig 7H), resulting in co-localization of DYRK1A with FAM53C in the cytoplasm (Fig 7H and I), again indicating the cytoplasmic anchoring function of FAM53C toward DYRK1A. In same microscopic fields, we often observed that DYRK1A was excluded from the nucleus of cells (*green*), where FAM53C expression levels (*magenta*) were high (Fig 7J and K, *arrows*). On the other hand, DYRK1A remained in the nucleus of cells where only low or no FAM53C was expressed (Fig 7J and K, *arrow heads*). This relationship between FAM53C expression levels and DYRK1A localization shows that FAM53C regulates intracellular

distribution of DYRK1A. Taken together, these results indicate that FAM53C binding suppresses nuclear accumulation of DYRK1A in cells and anchors DYRK1A in the cytoplasm.

**Tethering function of DYRK1A**

We have previously reported that DYRK1A binds to DCAF7/WDR68 and induces its nuclear accumulation (Miyata & Nishida, 2011). We then examined if FAM53C modifies the DYRK1A-induced nuclear localization of DCAF7/WDR68. When expressed alone, GFP-DCAF7/WDR68 localized both in the cytoplasm and nucleus (Fig 8A, *green*). Co-expression of HA-DYRK1A which accumulated in the nucleus (Fig 8B, *magenta*) induced nuclear co-localization of DCAF7/WDR68 with DYRK1A (Fig 8B, *green*). Additional expression of 3xFLAG-FAM53C resulted in cytoplasmic re-localization of DYRK1A (Fig 8D, *magenta*), as observed in Fig 7, and therefore, DYRK1A lost its ability to induce nuclear re-localization of DCAF7/WDR68 (Fig 8D, *green*). Expression of 3xFLAG-FAM53C (Fig 8C, *blue*) in the absence of DYRK1A did not influence the cellular localization of DCAF7/WDR68 (Fig 8C, *green*),

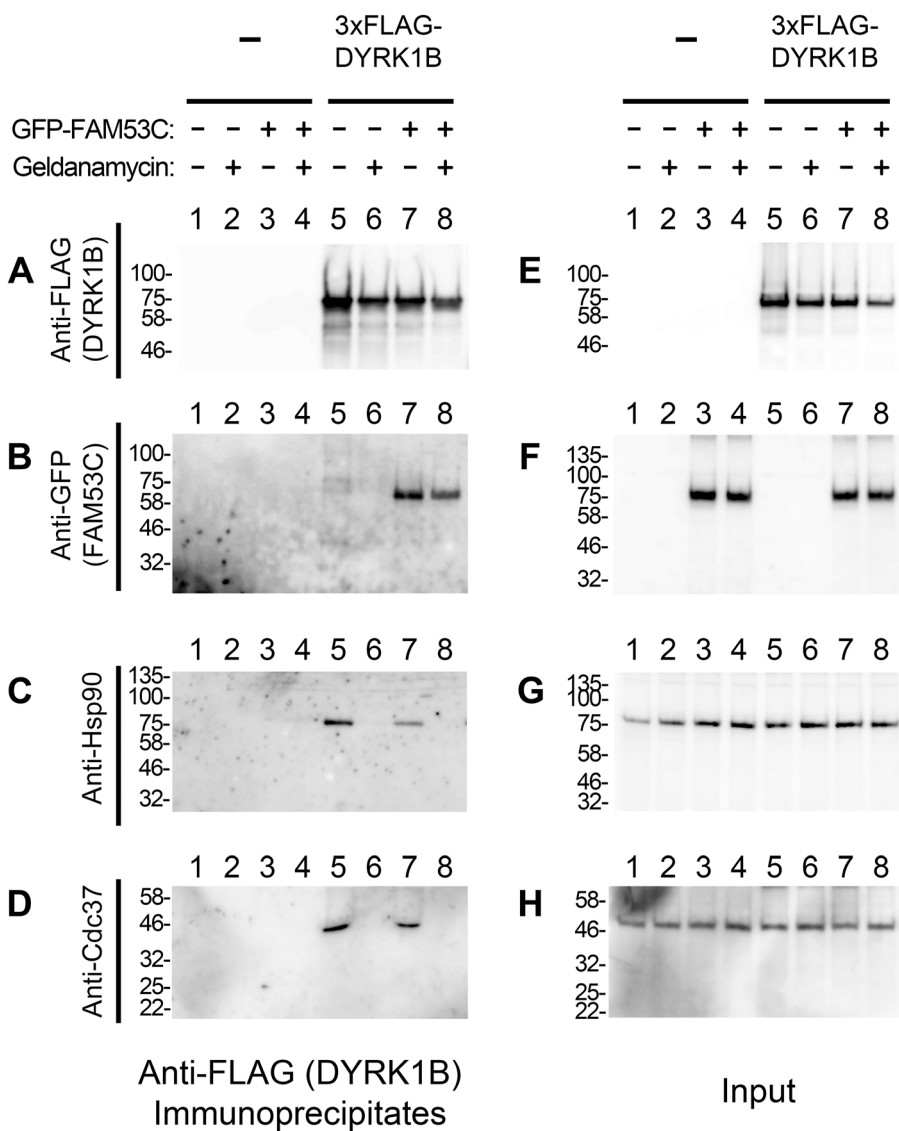

**Figure 9. Binding of FAM53C and Hsp90/Cdc37 to DYRK1B.**

3xFLAG-DYRK1B (*lanes 5–8*) was expressed in COS7 cells with (*lanes 3, 4, 7, and 8*) or without (*lanes 1, 2, 5, and 6*) GFP-FAM53C. Cells were treated with (*lanes 2, 4, 6, and 8*) or without (vehicle DMSO, *lanes 1, 3, 5, and 7*) Geldanamycin (2.5 μM 4 h), and the binding of FAM53C, Hsp90, and Cdc37 to DYRK1B was examined by co-immunoprecipitation experiments. **(A)** The amounts of immunoprecipitated DYRK1B were shown by Western blotting with anti-FLAG antibody. **(B)** The amounts of FAM53C co-immunoprecipitated with DYRK1B were shown by Western blotting with anti-GFP antibody. **(C)** The amounts of Hsp90 co-immunoprecipitated with DYRK1B were shown by Western blotting with anti-Hsp90 antibody. **(D)** The amounts of Cdc37 co-immunoprecipitated with DYRK1B were shown by Western blotting with anti-Cdc37 antibody. **(E, F, G, H)** The levels of indicated proteins in the cell extracts were examined by Western blotting. **(E)** FLAG-DYRK1B. **(F)** GFP-FAM53C. **(G)** Hsp90. **(H)** Cdc37.

which agrees with the observation that FAM53C did not directly bind to DCAF7/WDR68 in the co-immunoprecipitation assays (Fig 6). The same conclusion was obtained when we switched the tags. 3xFLAG-DCAF7/WDR68, when expressed alone, localized both in the cytoplasm and the nucleus (Fig 8E, *blue*), and co-expression of HA-DYRK1A (accumulated in the nucleus as shown in Fig 8F, *magenta*) induced nuclear co-localization of DCAF7/WDR68 (Fig 8F, blue) with DYRK1A. Expression of GFP-FAM53C (Fig 8G, *green*) in the absence of DYRK1A did not influence the cellular localization of DCAF7/WDR68 (Fig 8G, *blue*). Concurrent expression of GFP-FAM53C resulted in cytoplasmic co-localization of both DYRK1A (Fig 8H, *magenta*) and DCAF7/WDR68 (Fig 8H, *blue*) with FAM53C (Fig 8H, *green*). This result again indicates that FAM53C abrogates the ability of DYRK1A to anchor DCAF7/WDR68 in the nucleus. Taken altogether, it is concluded that DYRK1A tethers DCAF7/WDR68 and FAM53C by binding both of them, and that FAM53C anchors DYRK1A and DYRK1A-associated DCAF7/WDR68 in the cytoplasm.

**Binding of FAM53C and Hsp90/Cdc37 to DYRK1B**

The amino acid sequence identity is highest (85%) in the protein kinase catalytic domains of DYRK1A and DYRK1B, and FAM53C binds to the protein kinase domain of DYRK1A (Fig 4). We have previously shown that DYRK1B (but not DYRK1A) makes a stable complex with cellular molecular chaperone Hsp90 and its co-chaperone Cdc37 (Miyata & Nishida, 2021), and the Hsp90/Cdc37 chaperone system has been implicated in the maturation process of DYRK1B (Abu Jhaisha et al, 2017; Papenfuss et al, 2022). Hsp90/Cdc37 recognizes the catalytic kinase domains of various protein kinases; thus, we next examined the mutual relationship of DYRK1B-binding between Hsp90/Cdc37 and FAM53C. The binding of FAM53C to DYRK1B was examined by co-immunoprecipitation experiments in the presence or absence of an Hsp90-specific inhibitor Geldanamycin. Treatment of cells with Geldanamycin abolished the binding of both Hsp90 (Fig 9C, compare *lanes 5 & 6, lanes 7 & 8*) and Cdc37

(Fig 9D, compare *lanes 5 & 6, lanes 7 & 8*) to DYRK1B, in agreement with our previous investigation (Miyata & Nishida, 2021). Even after the complete disruption of Hsp90/Cdc37-DYRK1B binding, the amount of DYRK1B-associated FAM53C was not affected (Fig 9B, compare *lanes 7 & 8*). In addition, overexpression of FAM53C did not disrupt the binding of Hsp90 and Cdc37 to DYRK1B (Fig 9C and D, *lane 7*). We observed less Hsp90 and Cdc37 in the DYRK1B complexes from FAM53C-overexpressing cells; however, this may be because of the decreased levels of expressed and immunoprecipitated DYRK1B by the dual expression with FAM53C (Fig 9A and E, compare *lanes 5 and 7*), and quantification and normalization with DYRK1B levels indicated that the amounts of associated Hsp90/Cdc37 per DYRK1B was not significantly affected by the overexpression of FAM53C. The amounts of GFP-FAM53C, Hsp90, and Cdc37 in the cell extracts were shown in Fig 9F–H, respectively. These results indicate that Hsp90/Cdc37 binding is not required for the FAM53C binding to DYRK1B and that Hsp90/Cdc37 does not compete with FAM53C for binding to DYRK1B. The binding of Hsp90/Cdc37 and FAM53C may require different parts in the protein kinase domain of DYRK1B and is mutually independent.

## Discussion

### Tethering function and the protein interaction network of DYRK1A

DCAF7/WDR68 is the primary binding partner for DYRK1A (Miyata & Nishida, 2011; Glenewinkel et al, 2016; Yu et al, 2019). Our phospho-proteomic analysis identified FAM53C as a protein that makes a complex with DCAF7/WDR68 (Miyata & Nishida, 2021). Several other proteomic approaches have identified FAM53C as a DYRK1A inter-actor (Varjosalo et al, 2013; Guard et al, 2019; Menon et al, 2019; Roewenstrunk et al, 2019; Viard et al, 2022). These results indicate that FAM53C associates in cells with DYRK1A and/or DCAF7/WDR68. In this study, we directly show that DYRK1A binds to both FAM53C and DCAF7/WDR68 simultaneously with its different regions on the molecule. Whereas FAM53C is bound to the protein kinase domain of DYRK1A (Fig 4), DCAF7/WDR68 binds to the N-terminal domain of DYRK1A (Miyata & Nishida, 2011; Glenewinkel et al, 2016). DCAF7/WDR68 did not bind directly to FAM53C, but DYRK1A induced the association between DCAF7/WDR68 and FAM53C by binding to both of them, forming the DCAF7/WDR68-DYRK1A-FAM53C tri-protein complex. This result therefore indicates that our identification of FAM53C in the affinity-purified DCAF7/WDR68 complex should be because of the binding of FAM53C to DCAF7/WDR68 through en-dogenous DYRK1A in cells.

DYRK1A has been shown to associate in cells through DCAF7/WDR68 with several proteins including MEKK1 (Ritterhoff et al, 2010), adenovirus E1A (Glenewinkel et al, 2016), and IRS1 (Frendo-Cumbo et al, 2022). In these cases, DCAF7/WDR68 works as a scaffold to stimulate the protein-protein interaction between otherwise non-interacting partners. DCAF7/WDR68 is a WD40-repeat protein with a ring structure consisted of seven beta-propellers, and proteins with this structure often function as bases for protein-protein interactions (Stirnimann et al, 2010). On the other hand, our findings in this study indicate that DYRK1A can also work as a scaffold to

facilitate protein-protein interactions. Therefore, the DYRK1A-DCAF7/WDR68 pair assembles many proteins using both DYRK1A and DCAF7/WDR68, making this complex an important and efficient hub for many protein-protein interactions. Elucidation of the whole DYRK1A protein network should be fundamental for understanding the physiological function of DYRK1A in neurodevelopment and neu-rofunction at the molecular level.

We and others previously showed that DYRK1B, but not DYRK1A, makes stable complex with a set of molecular chaperones, in-cluding Hsp90, its co-chaperone Cdc37, and Hsp70 (Abu Jhaisha et al, 2017; Miyata & Nishida, 2021; Papenfuss et al, 2022). Hsp90/Cdc37 recognizes the catalytic domains of client protein kinases, as in the case of FAM53C; thus, the relationship between the binding of FAM53C and Hsp90/Cdc37 matters. Theoretically, FAM53C may bind to DYRK1B through Hsp90/Cdc37; however, this possibility may be unlikely because DYRK1A, which does not make a stable complex with Hsp90/Cdc37, still binds to FAM53C. In addition, the complete dissociation of Hsp90/Cdc37 from DYRK1B by the Geldanamycin treatment did not abolish nor enhance the binding of FAM53C to DYRK1B (Fig 9), indicating that the Hsp90 binding to DYRK1B is dispensable and permissive for the FAM53C binding.

### FAM53C phosphorylation

Our phospho-proteomic analysis identified nine phosphorylation sites in FAM53C (Fig 1A), and all of them are on serines (Ser86, Ser122, Ser162, Ser234, Ser247, Ser255, Ser273, Ser299, and Ser350). Results by high-throughput phospho-proteomic analyses in the PhosphoSite database (https://www.phosphosite.org/proteinAction.action?id=6364) indicate that all the phosphoserines identified in this study, except pSer-350, can be observed in human, mouse, and rat. In ad-dition, the PhosphoSite database indicates that pSer122 and pSer162 are sensitive to Torin1 (an inhibitor for mTOR) and AZD1152/ZM447439 (inhibitors for Aurora kinase)/BI2536 (an inhibitor for PLK1), respectively. It remains unclear if FAM53C is a direct substrate of these kinases in cells. Four of the identified phosphorylation sites (pSer86, pSer162, pSer234, and pSer255) are immediately followed by a proline, suggesting that these sites of FAM53C might be phosphorylated by certain proline-directed protein kinases. Among them, the amino acid se-quence surrounding pSer86 (RGNpSPKE) matches the DYRK1A substrate consensus sequence (RXXpSP) (Himpel et al, 2000; Aranda et al, 2011). The amino acid sequences surrounding pSer122 (RSLpSVP) and pSer273 (RSRpSQP) are consistent with the consensus sequence motif (RXXpSXP) for phospho-dependent 14-3-3 binding (Pennington et al, 2018). In fact, the BioPlex interactome database (Fig 1B, *lower left panel*) suggests interactions of FAM53C with several 14-3-3 proteins including YWHAB(14-3-3β), YWHAG(14-3-3γ), YWHAH(14-3-3η), YQHAQ(14-3-3θ), and YWHAZ(14-3-3ζ). 14-3-3 proteins interact with numerous structurally and functionally diverse targets and act as central hubs of cellular signaling networks (Pennington et al, 2018). DYRK1A may therefore participate in wide varieties of 14-3-3 dependent cellular signaling pathways through FAM53C binding. A very recent proteomic interactome study with all 14-3-3 human paralogs shows interaction of DYRK1A and FAM53C

with five and six members of seven 14-3-3 proteins (Segal et al, 2023), respectively.

### FAM53C may function to keep DYRK1A in a kinase-inactive state in the cytoplasm

Triplication of *DYRK1A* gene is responsible for many pathological phenotypes observed in Down syndrome patients. Therefore, the regulation of DYRK1A function is of physiological and clinical importance. Many low-molecular-weight compounds have been developed in the past decade as specific DYRK1A inhibitors (Duchon & Hérault, 2016; Stotani et al, 2016; Feki & Hibaoui, 2018; Arbones et al, 2019; Kumar et al, 2021); however, only few endogenous proteins such as RanBPM and SPREAD (Zou et al, 2003; Li et al, 2010) have been proposed to work inhibitory to DYRK1A. Here in this study, we revealed that FAM53C suppresses the protein kinase activity of DYRK1A by binding to its catalytic domain. The suppression was observed both in phosphorylation of a well-established DYRK1A substrate MAPT/Tau and in autophosphorylation of DYRK1A itself. DYRK1A-dependent MAPT/Tau phosphorylation is one of the molecular bases for the early onset of Alzheimer disease in most Down syndrome patients (Ryoo et al, 2007), and DYRK1A inhibition is believed to be effective for treatment of Alzheimer disease (Stotani et al, 2016; Branca et al, 2017). FAM53C, by suppressing DYRK1A activity, may possibly also be involved in Alzheimer disease. DYRK1A is known to autophosphorylate on a tyrosine residue in the activation loop, and this autophosphorylation is required for its maturation and full activity (Himpel et al, 2001; Lochhead et al, 2005). However, the autophosphorylation shown in this study is different from the tyrosine autophosphorylation because the tyrosine autophosphorylation is a one-off event during the translational process, and the FAM53C-sensitive autophosphorylation was observed in a post-maturation stage during the incubation of affinity-purified DYRK1A with $Mg^{2+}$-ATP in vitro. The binding of FAM53C may interfere with the access of ATP or substrates to DYRK1A or inactivate DYRK1A kinase by inducing its conformational alteration. Structural analysis of the FAM53C-DYRK1A complex may shed light on the molecular mechanism of the DYRK1A inhibition by FAM53C.

DYRK1A encodes a bipartite NLS in the N-terminal domain and accumulates in the nucleus when exogenously overexpressed in various cell lines (Becker et al, 1998; Álvarez et al, 2003; Miyata & Nishida, 2011). However, many studies with various antibodies against DYRK1A have indicated that endogenous DYRK1A resides within the cytoplasm of brain tissues and in cell lines (Martí et al, 2003; Wegiel et al, 2004; Ferrer et al, 2005; Aranda et al, 2008; Nguyen et al, 2018). This discrepancy has suggested that there may be an unveiled molecular mechanism responsible for anchoring DYRK1A in the cytoplasm. DYRK1A regulates gene expression by phosphorylating nuclear substrates. For example, DYRK1A-dependent phosphorylation of a transcription factor NFAT, which regulates immuno-responsive, inflammatory, and developmental processes, induces its cytoplasmic re-localization (Arron et al, 2006; Gwack et al, 2006). DYRK1A is also involved in transcriptional regulation by interacting with histone acetyl transferase p300 and CBP (Li et al, 2018) and by phosphorylating the C-terminal domain repeat of RNA polymerase II (Yu et al, 2019). Our results indicated that FAM53C

binds to DYRK1A and keeps DYRK1A inactive in the cytoplasm, suggesting that FAM53C prevents DYRK1A from phosphorylating nuclear substrates. FAM53C is localized in the cytoplasm, and as the amount of FAM53C in cells increases, DYRK1A remains in the cytoplasm with FAM53C to a greater extent (Fig 7). In the endogenous situation in vivo, DYRK1A and FAM53C levels remain in physiological balance; therefore, FAM53C can finely regulate the intracellular distribution of DYRK1A, appropriately keeping DYRK1A in the cytoplasm until needed. If DYRK1A is overexpressed beyond the available endogenous FAM53C level, the excess FAM53C-free DYRK1A may translocate into the cell nucleus and aberrantly phosphorylate nuclear proteins, leading to modified gene expression (Fig 10). This may be one of the reasons why a mere 1.5-fold overexpression of DYRK1A can induce drastic changes in gene expression and pathophysiological consequences in Down syndrome patients. *FAM53C* knock-out mice do not reproduce all the *DYRK1A* trisomy phenotypes, indicating the involvement of other molecular mechanisms, such as post-translational modifications or the binding of other interacting partners, in the regulation of DYRK1A activity and localization. In addition, our findings are based on the experiments conducted with overexpressed immortalized cell lines; therefore, the observed results may not precisely represent the role of FAM53C in the impact of the DYRK1A dosage effect in Down syndrome.

In conclusion, this study identified FAM53C as a binding protein of DYRK1A. DYRK1A is concurrently bound to both FAM53C and DCAF7/WDR68 via its kinase domain and N-terminal domain, respectively, forming a tri-protein complex. FAM53C binding suppressed the protein kinase activity of DYRK1A towards MAPT/Tau and its own autophosphorylation, anchoring the DYRK1A-DCAF7/WDR68 complex in the cytoplasm in an inactive state. The amount of FAM53C in cells determined the appropriate intracellular distribution of DYRK1A and DCAF7/WDR68. In addition, FAM53C is bound to DYRK1B in an Hsp90/Cdc37-independent manner. These results explain in part why endogenous DYRK1A in animal brain tissues is often observed in the cytoplasm despite the strong tendency of DYRK1A nuclear localization. The identification of FAM53C as a suppressive binding partner for DYRK1A sheds new light on the molecular mechanism of Down syndrome caused by triplication of *DYRK1A* in human chromosome 21.

# Materials and Methods

### Reagents and antibodies

An antibody specific for FAM53C was raised in a rabbit against a KLH-conjugated peptide, CQQDFGDLDLNLIEEN, corresponding to amino acids 377–392 (C-terminal end) of orangutan FAM53C. The antiserum was purified on an affinity column with the antigen-peptide-conjugated resin. The peptide synthesis, immunization, and affinity purification were performed at Cosmo Bio Co., Ltd. Anti-FLAG antibody (M2), anti-FLAG (M2)-affinity resin, 3xFLAG peptide, and 3xFLAG-CMV7.1 vector were from Sigma-Aldrich, anti-GFP antibody (JL8) was from Clontech, and anti-Cdc37 antibody (E-4) was from SantaCruz. Anti-HA antibodies were from Roche (clone 12CA5 for Western blotting)

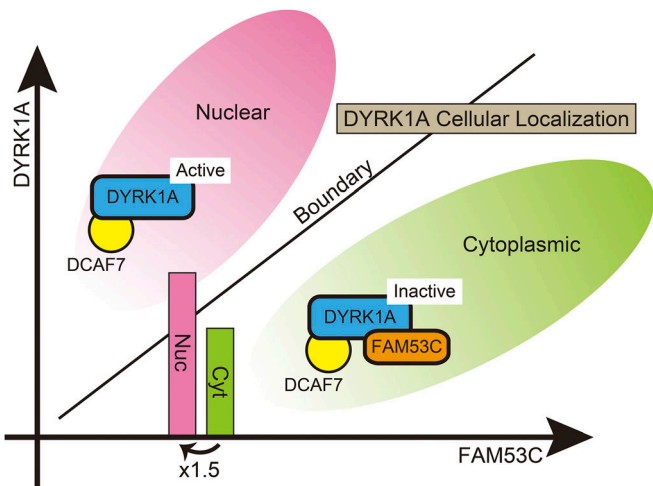

**Figure 10. Schematic illustrations of the suppressive anchoring function of FAM53C.**
Regulation of the intracellular localization of DYRK1A by a balance between levels of DYRK1A and FAM53C. Sufficient amounts of FAM53C are required for the efficient anchoring of DYRK1A in the cytoplasm in an inactive state (*green area*). FAM53C-free DYRK1A is active and translocates into the cell nucleus with DCAF7/WDR68 (*red area*). In certain conditions, a 1.5-times increase of DYRK1A (shown by bars), by crossing over the balance threshold (shown by a boundary line), might induce a drastic change of intracellular distribution of DYRK1A from the cytoplasm (*green area*) to the nucleus (*red area*).

or from SantaCruz (clone Y11 for immunofluorescent staining). Anti-Hsp90 antibody (Koyasu et al, 1986) and anti-DCAF7/WDR68 antibody (Miyata & Nishida, 2011) were described before. Geldanamycin was purchased from Life Technologies Inc. or from InvivoGen, and stock solution was prepared at 5 mM in DMSO. pcDNA3-HA and pEGFP-C1Not vectors were previously described (Miyata et al, 1999; Miyata & Nishida, 2011, 2021). Hoechst 33342 was from Molecular Probes.

### Isolation of human FAM53C cDNA

A cDNA fragment encoding human FAM53C was isolated by amplifying with nested PCR from human cDNA library plasmid (Takara). The oligonucleotide primer sequences used for the first PCR are 5′-CAAAGTGTGCAAGTCAAATCCTGG-3′ (5′ upstream) and 5′-CGGCTGGTTCTTTCCGCCTC-3′ (antisense primer in the vector region). The oligonucleotide primer sequences used for the second PCR are 5′-GCGGCCGCTATGATAACCCTGATCACTGAG-3′ (*Not* I + first Met) and 5′-GCGGCCGCTTAGTTTTCCTCAATCAAATTC-3′ (3′ end + *Not* I, antisense). As a result, the amplified PCR fragment of FAM53C coding region contains a *Not* I site immediately 5′ upstream of the starting ATG and another *Not* I site 3′ downstream of the stop codon. The obtained PCR fragment was inserted into pCR2.1 Topo vector (Invitrogen), and the whole coding region was verified by direct sequencing. The *Not* I fragment encoding the entire FAM53C coding region was purified by low-melting gel electrophoresis. In the NCBI nucleotide database, three transcription variants are found for human FAM53C, and we obtained isoform 1 (392 amino acids) with our cloning strategy.

### Mammalian expression vectors for DYRK1A, DYRK1B, and FAM53C

Expression plasmids for 3xFLAG-tagged DYRK1A(WT), DYRK1A(N), DYRK1A(K), DYRK1A(C), DYRK1A(N+K), DYRK1A(K+C), and DYRK1B(WT) were previously described (Miyata & Nishida, 2011, 2021). *Not* I fragments encoding the N-terminal (N), the kinase domain (K), and the N-terminal domain+the kinase domain (N+K) of DYRK1A were ligated into the *Not* I site of pEGFP-C1Not to obtain plasmids of GFP-fusion proteins for DYRK1A(N), DYRK1A(K), and DYRK1A(N+K), respectively. HA-DYRK1A expression plasmid was previously described (Miyata & Nishida, 2011). The *Not* I fragment of FAM53C was ligated into the *Not* I site of p3xFLAG-CMV7.1 or pEGFP-C1Not to obtain plasmids for expression of 3xFLAG-tagged or GFP-tagged FAM53C.

### Expression in mammalian cells and immunoprecipitation experiments

COS7 and NIH-3T3 cells were cultured in DMEM supplemented with 10% FCS at 37°C. Cells were transfected with mammalian expression vectors by electroporation as described previously (Miyata et al, 1997, 1999), or by lipofection with Lipofectamine LTX plus or with Lipofectamine 2000 according to the protocol supplied by the manufacture. Cell extracts were prepared as described before (Miyata et al, 1997, 1999). Extracts with equal amounts of proteins were incubated with 10 μl of anti-FLAG resin for 12 h at 4°C. The immunocomplexes were washed, isolated, and FLAG-tagged proteins were eluted and analyzed, as described previously (Miyata et al, 2001; Miyata & Nishida, 2004).

### Immunofluorescent staining

COS7 and NIH-3T3 cells were transfected with plasmids encoding GFP-, 3xFLAG-, or HA-tagged proteins and cultured in 35-mm dishes on glass cover slips. 24 h later, cells were fixed with 10% formaldehyde (37°C 20 min), stained with anti-FLAG- or anti-HA antibody, and observed with a fluorescent microscope (Axiophot, Zeiss, or IX71, Olympus) essentially as described (Miyata & Nishida, 2004, 2021). For staining of the nuclei, cells were incubated with 1 μg/ml of Hoechst 33342, 3% BSA, 0.1% goat IgG in PBS at room temperature for 60 min. All the immunofluorescent staining experiments were repeated with consistent results, and typical representative cell images are shown.

### In vitro DYRK1A protein kinase assay

The protein kinase activity of DYRK1A was determined in vitro with recombinant MAPT/Tau protein as an exogenous substrate. A DNA fragment encoding human full length hT40 MAPT/Tau was obtained by PCR with plasmid DNA encoding WT full-length hT40 MAPT/Tau as a template. The oligonucleotide primer sequences used are 5′-GCGGCCGCCATGGCTGAGCCCCGCCAGGAG-3′ (*Not* I + first Met) and 5′-GCGGCCGCTCACAAACCCTGCTTGGCCAG-3′ (3′ end + *Not* I, antisense). As a result, the amplified PCR fragment of MAPT/Tau coding region contains a *Not* I site immediately 5′ upstream of the starting ATG and another *Not* I site 3′ downstream of the stop codon. The *Not* I fragment was verified by direct sequencing and inserted into the *Not* I site of pGEX6P2 (Cytiva), and the expression vector was introduced into an *Escherichia coli* strain BL21-CodonPlus (DE3)-RIL

(Agilent). The transformed bacterial cells were grown at 37°C until $OD_{600}$ reached 0.7, and isopropyl-$\beta$-D-thiogalactopyranoside was added at a final concentration of 0.5 mM to induce the expression of GST-MAPT/Tau followed by 2-h incubation at 22°C. The bacterial cells were collected by centrifugation (7,000$g$ 15 min 2°C), washed once with cold PBS, frozen at –80°C, and solubilized in B-PER solution (Pierce) supplemented with 1/100 (vol/vol) of bacterial protease inhibitor cocktail (Sigma-Aldrich), 1 mM EDTA, and 0.5 M NaCl. The extract was clarified by a centrifugation (18,000$g$ 15 min 2°C) and mixed with glutathione-Sepharose (Cytiva) for 4 h followed by three washes with PBS + TritonX100 (1%) and three washes with cleavage buffer (50 mM Tris, 150 mM NaCl, 1 mM EDTA, 1 mM DTT, pH 7.0). The GST moiety of the fusion protein was removed by incubating with PreScission protease (Cytiva) for 12 h at 4°C, and released MAPT/Tau protein was purified with HiTrap CM Sepharose FF (Cytiva) with a NaCl gradient (50–1,000 mM) in purification buffer (50 mM MES, 1 mM DTT, 1 mM EDTA, pH 6.4). Eluted fractions containing MAPT/Tau protein were collected, desalted with a PD10 column (Cytiva) to 50 mM NaCl in the purification buffer, and concentrated by an Amicon Ultra-15 filter unit (30,000 D cut-off; Millipore). The recombinant MAPT/Tau protein was >98% pure as revealed by CBB staining.

3xFLAG-DYRK1A expressed in COS7 cells was affinity purified as described above and incubated with purified recombinant MAPT/Tau protein for 30 min at 30°C with gentle shaking for phosphorylation reactions. Composition of the reaction mixture (20 $\mu$l) was 50 mM Hepes, 10 mM Tris, 1.25 mM MES, 81.25 mM NaCl, 2% glycerol, 10 mM $MgCl_2$, 5 mM ATP, 0.2% NP40, 0.225 mM DTT, 0.425 mM EDTA, pH 7.4, containing 0.35 $\mu$g (17.5 $\mu$g/ml) MAPT/Tau. The reactions were stopped by adding SDS-sample buffer, followed by an incubation at 98°C for 5 min. DYRK1A-dependent phosphorylation of Thr212 of MAPT/Tau protein was evaluated by Western blotting with anti-phospho Tau (Thr212) antibody (#44740G; Invitrogen).

### Other procedures

SDS–PAGE was performed with 8% or 10% acrylamide gels and BOLT MES (Invitrogen) or Tris-glycine SDS running buffer. Western blotting was performed using polyvinylidene difluoride filter membranes (Millipore) for electro-transfer and Blocking One (Nacalai Tesque) for blocking. Signals were developed with horseradish peroxidase-conjugated secondary antibodies (GE Healthcare Bio-Sciences) or peroxidase-conjugated primary antibodies using Western Lightning Plus-ECL (PerkinElmer) and the chemiluminescent system AI680 (GE Healthcare) as described (Miyata & Nishida, 2004). In some experiments, polyvinylidene difluoride membrane wetting was performed with ethanol instead of methanol, and we confirmed that this does not affect the results. For all the Western blotting data, we repeated independent experiments and showed representative images of obtained consistent results.

## Data Availability

This study does not contain deposited data in external repositories. Original data can be provided upon request.

## Supplementary Information

## Acknowledgements

We thank T Sakabe, T Aoki, and M Nakagawa for their excellent technical assistance. We thank Drs. B Chambraud and EE Baulieu (INSERM U1195, France) for providing us plasmid DNA encoding human hT40 MAPT/Tau. This work was supported by Grants-in-Aid for Scientific Research from the Ministry of Education, Culture, Sports, Science and Technology of Japan (JSPS KAKENHI Grant Numbers JP25440046, JP18K06131).

### Author Contributions

Y Miyata: conceptualization, resources, data curation, formal analysis, funding acquisition, validation, investigation, visualization, methodology, and writing—original draft, review, and editing.
E Nishida: resources, supervision, funding acquisition, and writing—review and editing.

### Conflict of Interest Statement

The authors declare that they have no conflict of interest.

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
