## [Reviewer comments · Life Science Alliance]

Life Science Alliance

Identification of FAM53C as a cytosolic-anchoring inhibitory binding protein of the kinase DYRK1A

Yoshihiko Miyata and Eisuke Nishida

DOI: <https://doi.org/10.26508/lsa.202302129>

Corresponding author(s): Yoshihiko Miyata, Kyoto University

Review Timeline:

Submission Date:	2023-05-03
Editorial Decision:	2023-05-25
Revision Received:	2023-08-18
Editorial Decision:	2023-09-19
Revision Received:	2023-09-26
Accepted:	2023-09-26

Transaction Report:

May 25, 2023

Re: Life Science Alliance manuscript #LSA-2023-02129-T

Dr. Yoshihiko Miyata
Kyoto University
Graduate School of Biostudies, Department of Cell and Developmental Biology
Yoshida Hon-machi
Room 304, Research Building No.16
Sakyo-ku, Kyoto 606-8501
Japan

Dear Dr. Miyata,

Thank you for submitting your manuscript entitled "Identification of FAM53C as a suppressive binding protein of a neurodevelopmental disorders-related kinase DYRK1A" to Life Science Alliance. The manuscript was assessed by expert reviewers, whose comments are appended to this letter. We invite you to submit a revised manuscript addressing the Reviewer comments.

Thank you for this interesting contribution to Life Science Alliance. We are looking forward to receiving your revised manuscript.

Sincerely,

B. MANUSCRIPT ORGANIZATION AND FORMATTING:

Reviewer #1 (Comments to the Authors (Required)):

The authors have mechanistically characterized FAM53C as a new DYRK1-interacting protein with unique properties. Most importantly, they demonstrate that FAM53C binds to the catalytic domain of DYRK1A and strongly reduces the kinase activity, and that FAM53C anchors DYRK1, together with its binding partner DCAF7, in the cytoplasm, thereby inhibiting the nuclear functions of this protein kinase. These results represent not only the very first functional characterization of FAM53C, which has not yet been studied in a single scientific publication, but also provide interesting clues about the cellular mechanism which may play an important role in the regulation of the protein kinases and DYRK1B. The discussion offers an inspiring hypothesis to explain the marked gene dosage effect of DYRK1A that underlies its role in Down syndrome. Overall, the study is well conducted and supports the conclusions drawn by the authors. The mechanistic experiments rely on overexpressed proteins, but the fact that FAM53C indeed associates with DYRK1A at endogenous expression levels has already been repeatedly shown in DYRK1A interactome screens. The experiments that characterize the binding events that underlie the FAM53C/DYRK1/DCAF7 complex are convincing (Fig. 2-6), as well as the effect on DYRK1A activity (Fig. 5) and the subcellular localization experiments (Fig. 7-8). The manuscript is well organized and understandable, in spite of considerable language errors and uncommon phrasing. Several paragraphs are verbose and lengthy and merit focusing. I recommend language improvement.

Specific comments:

Title:

"Identification of FAM53C as a suppressive binding protein of a neurodevelopmental disorders-related kinase DYRK1A"

The term "suppressive binding protein" can only be understood after reading of the report and is not helpful in the title. It is not clear what is suppressed by FAM53C?

Introduction:

In my opinion, the general part on the importance of DYRK1A physiology and pathology has a review-like character rather than introducing into the specific research objective of this study. e.g., 6 references are cited just to support the role of DYRK1A in diabetes mellitus. I suggest to condense this part and delete many of the references and maybe rather include the information that DYRK1A has not only nuclear but also cytoplasmic substrates. Among many other examples, mitochondrial TOM70 is considered an important substrate of DYRK1A, which should be considered when discussing the role of FAM53C as a regulator of the balance between nuclear and cytoplasmic DYRK1A.

Figure 1, page 5, page 13. This figure provides minimal useful or new information but illustrates data from other papers and databases. Fig 1 shows that FAM53C was already identified - but not verified - as interaction partner of DYRK1A and that it is predicted to be an unstructured protein with a few phosphorylation sites. The paragraph on the Bioplex databases (page 5) is lengthy. The fact that FAM53C/DCAF7 and both DYRK1A and DYRK1B were identified as binding proteins can be stated in a one sentence.

(minor comment: the alphafold color code missing in the panel C, in contrast to the text no "high" confidence structural element is present).

Fig. 1A, page 13: The discussion of the potential phosphorylation of FAM53C by DYRK1A does not interpret results of the present study, not really informative and rather speculative. Fig 1A shows results from a previous study, and it is not clear to me whether or how the observed phosphosites are related to the DYRK1A or are dependent on DYRK1A. There are no kinase assays in vitro or in cells other the TAU assay in the present study. Given that there seems to be no further experimental evidence, the discussion can be limited to a comment that FAM53C may be phosphorylated by DYRK1, maybe with a speculative comment that this could promote 14-3-3 interaction (with unknown consequences, however).

In Fig8, DCAF7/WDR68 is just labeled "WDR68". Generally the authors use both names of the protein in the text and in the labelling of the figures, which is an acceptable decision. When only one designation is used, it should be the official gene name, i.e. DCAF7.

Fig 10 A is overdone, a small scheme like in Fig 6 would be sufficient

Language: There are significant language problems, e.g. use of articles.

Here are selected examples from the abstract:

"A protein kinase" The indefinite article is not correct

"encoded in" should read "encoded on"

"major contributor for" should read "major contributor to"

"the protein kinase activity of DYRK1A to itself" autophosphorylation activity

"FAM53C is thus a binding suppressor of DYRK1A" "binding suppressor" is not an established term

"in the normal brain tissues" should read "in normal brain tissue"

"gene expression modification caused by DYRK1A" "regulation" rather than modification sound more appropriate)

page 13: "trisomization" this term is quite unusual and should not be used for genes (but for chromosomes)

Introduction:

"pleiotropic substrates" do the authors really want to say that the substrates are pleiotropic?

"in our brain" in human brain

Discussion (1 sentence) "most major binding partner"

"Two other proteomic approaches have suggested FAM53C as a DYRK1A-interactor" "identified" would be more appropriate than "suggested".

Reviewer #2 (Comments to the Authors (Required)):

The current manuscript shed light on the interaction of FAM53C with DYRK1A and DCAF7/WDR68 and proposed the regulatory function of the interaction by controlling DYRK1A localization in the cytoplasm. It also gives some information about interaction with DYRK1B.

The experiments are well described and supported the main conclusion about the binding of FAM53C to the Kinase domain of DYRK1A regulating its kinase effect on TAU/MAPT. Some questions are still open:

1) Are DYRK1A or 1B able to phosphorylate FAM53C?

2) all the cellular conclusions are based on overexpression in COS7 and NIH3T3 immortalized cells and thus should be tempered when speaking about dosage effect in Down syndrome. IN particular, accumulation of DYRK1A in brain neurons has been reported in the cytoplasm... Thus, this fact, reported by several publications, should be considered while discussing the role of FAM53C in DS.

Reviewer #1

We sincerely appreciate the insightful comments of the reviewer for our manuscript. We have significantly modified the text according to the suggestion of the reviewer as shown below:

> Title:

- > The term "suppressive binding protein" can only be understood after reading of the report and is not helpful in the title. It is not clear what is suppressed by FAM53C?

We replaced “suppressive binding protein” with “cytosolic-anchoring inhibitory binding protein”. Due to the character number limitation of the Journal, we deleted “a neurodevelopmental disorders-related” from the title. We believe that the new title now indicates more clearly that FAM53C suppresses the nuclear localization and kinase activity of DYRK1A.

> Introduction:

- > In my opinion, the general part on the importance of DYRK1A physiology and pathology has a review-like character rather than introducing into the specific research objective of this study. e.g., 6 references are cited just to support the role of DYRK1A in diabetes mellitus.
- > I suggest to condense this part and delete many of the references and maybe rather include the information that DYRK1A has not only nuclear but also cytoplasmic substrates. Among many other examples, mitochondrial TOM70 is considered an important substrate of DYRK1A, which should be considered when discussing the role of FAM53C as a regulator of the balance between nuclear and cytoplasmic DYRK1A.

We shortened the first section of the introduction and deleted many references there according to the reviewer’s suggestion. Two recent review articles are added instead. In addition, we added text and references emphasizing the cytoplasmic role of DYRK1A. In particular, we now refer to TOM70 as a cytoplasmic substrate of DYRK1A in the introduction, suggesting physiological importance of cytosolic function of DYRK1A. We deeply appreciate the reviewer for this informative suggestion. We believe that the revised introduction is more condensed, showing the specific research objective of this study more clearly.

- > Figure 1, page 5, page 13. This figure provides minimal useful or new information but
- > illustrates data from other papers and databases. Fig 1 shows that FAM53C was
- > already identified - but not verified - as interaction partner of DYRK1A and that it is
- > predicted to be an unstructured protein with a few phosphorylation sites.
- > The paragraph on the Bioplex databases (page 5) is lengthy. The fact that
- > FAM53C/DCAF7 and both DYRK1A and DYRK1B were identified as binding proteins
- > can be stated in a one sentence.

We agree with the reviewer that Figure 1 is based mostly on already obtained data except the newly-identified phosphorylation sites of FAM53C. As the reviewer wrote, this manuscript is the very first functional characterization of FAM53C, therefore, we believe it would be helpful to include the sequential and structural overview as well as the interaction network of FAM53C. According to the reviewer's suggestion, we have significantly shortened the paragraph on the Bioplex analysis.

- > minor comment: the alphafold color code missing in the panel C, in contrast to the text
- > no "high" confidence structural element is present).

The color code is now added in Figure 1C.

- > Fig. 1A, page 13: The discussion of the potential phosphorylation of FAM53C by
- > DYRK1A does not interpret results of the present study, not really informative and
- > rather speculative. Fig 1A shows results from a previous study, and it is not clear to
- > me whether or how the observed phosphosites are related to the DYRK1A or are
- > dependent on DYRK1A. There are no kinase assays in vitro or in cells other the TAU
- > assay in the present study. Given that there seems to be no further experimental
- > evidence, the discussion can be limited to a comment that FAM53C may be
- > phosphorylated by DYRK1, maybe with a speculative comment that this could promote
- > 14-3-3 interaction (with unknown consequences, however).

We agree with the reviewer that there is no clear evidence that shows direct phosphorylation of FAM53C with DYRK1A. We therefore deleted and modified sentences suggesting the phosphorylation of FAM53C by DYRK1A in the revised manuscript. We hope that we may examine DYRK1-dependent phosphorylation of FAM53C in our future studies.

- > In Fig8, DCAF7/WDR68 is just labeled "WDR68". Generally the authors use both
- > names of the protein in the text and in the labelling of the figures, which is an
- > acceptable decision. When only one designation is used, it should be the official gene
- > name, i.e. DCAF7.

We corrected Fig 8 and now all the labels are shown as DCAF7/WDR68.

- > Fig 10 A is overdone, a small scheme like in Fig 6 would be sufficient

We deleted Fig 10A in the revised manuscript according to the suggestion of the reviewer.

- > Language: There are significant language problems, e.g. use of articles.
- > Here are selected examples from the abstract:
- > "A protein kinase" The indefinite article is not correct
- > "encoded in" should read "encoded on"
- > "major contributor for" should read "major contributor to"
- > "the protein kinase activity of DYRK1A to itself" autophosphorylation activity
- > "FAM53C is thus a binding suppressor of DYRK1A" "binding suppressor" is not an
- > established term
- > "in the normal brain tissues" should read "in normal brain tissue"
- > "gene expression modification caused by DYRK1A" "regulation" rather than
- > modification sound more appropriate)
- > page 13: "trisomization" this term is quite unusual and should not be used for genes
- > (but for chromosomes)

> Introduction:

- > "pleiotropic substrates" do the authors really want to say that the substrates are
- > pleiotropic?
- > "in our brain" in human brain
- > Discussion (1 sentence) "most major binding partner"

- > "Two other proteomic approaches have suggested FAM53C as a DYRK1A-interactor"
- > "identified" would be more appropriate than "suggested".

We deeply appreciate the reviewer for pointing out language problems in our manuscript. We have corrected all the above points according to the suggestion of the reviewer.

Reviewer #2

> Some questions are still open:

> 1) Are DYRK1A or 1B able to phosphorylate FAM53C?

As pointed out by both the reviewers, this is one of major remaining subjects that we did not fully examine in this manuscript. We speculated that DYRK1A and DYRK1B may phosphorylate FAM53C based on the fact that one of the phosphorylation sites (Ser86) we have identified in FAM53C matches the DYRK1A substrate consensus sequence. However, we don't have experimental results showing direct phosphorylation of FAM53C by DYRK1A, so we deleted and modified speculative sentences in the revised manuscript according to the suggestion of reviewers.

To answer the question raised by the reviewer #2, we newly conducted several preliminary experiments. We co-expressed 3xFLAG-tagged DYRK1A or DYRK1B with GFP-tagged FAM53C in cultured cells and examined if DYRK1 expression induces a SDS-PAGE mobility shift (suggesting phosphorylation) of FAM53C. When the expression of DYRK1 is low (DYRK1A), the co-expression of FAM53C inhibited DYRK1 activity, therefore, FAM53C was not phosphorylated. This is in good agreement with our results shown in this manuscript that FAM53C inhibits DYRK1 activity (Figure 5). On the other hand, very high overexpression of DYRK1 (kinase-active version of DYRK1B) induced the mobility up-shift of FAM53C in a kinase-activity dependent manner, suggesting DYRK1-induced FAM53C phosphorylation in cells. This observation is in cultured cells, so there may be intermediating protein kinase(s) that phosphorylate(s) FAM53C in a DYRK1-dependent manner. To prove the direct phosphorylation of FAM53C in vitro with purified components, we have tried to obtain bacterially-expressed purified FAM53C. However, GST-FAM53C expressed in E.coli was observed to be highly degraded in an insoluble fraction even at lower culture temperatures with protease-deficient host cells, and we are still not able to get pure full length FAM53C after optimization trials. This is not surprising considering the predicted highly disordered property of FAM53C (Figure 1C). Therefore, we still have not yet revealed clearly if DYRK1 directly phosphorylates FAM53C. We hope that we can continue working on this subject in the future.

> 2) all the cellular conclusions are based on overexpression in COS7 and NIH3T3

- > immortalized cells and thus should be tempered when speaking about dosage effect in
- > Down syndrome. IN particular, accumulation of DYRK1A in brain neurons has been
- > reported in the cytoplasm... Thus, this fact, reported by several publications, should be
- > considered while discussing the role of FAM53C in DS.

We agree with the reviewer that our findings are based on the experiments conducted with overexpressed immortalized cell lines, thus, the obtained conclusions should not immediately be applied to natural Down syndrome brains. We added one sentence to clarify this point at the end of the discussion. In addition, we have included two additional references showing the accumulation of DYRK1A in brain neurons in the Introduction and Discussion according to the suggestion of the reviewer.

September 19, 2023

RE: Life Science Alliance Manuscript #LSA-2023-02129-TR

Dr. Yoshihiko Miyata
Kyoto University
Graduate School of Biostudies, Department of Cell and Developmental Biology
Yoshida Hon-machi
Room 304, Research Building No.16
Sakyo-ku, Kyoto 606-8501
Japan

Dear Dr. Miyata,

Thank you for submitting your revised manuscript entitled "Identification of FAM53C as a cytosolic-anchoring inhibitory binding protein of the kinase DYRK1A". We would be happy to publish your paper in Life Science Alliance pending final revisions necessary to meet our formatting guidelines.

- please add the Twitter handle of your host institute/organization as well as your own or/and one of the authors in our system
- please remove the Character count and Word count from the manuscript file
- please add a callout for Figure 8G to your main manuscript text

Figure checks:

- please include scale bars for Figures 7 and 8

A. FINAL FILES:

B. MANUSCRIPT ORGANIZATION AND FORMATTING:

Sincerely,

Reviewer #1 (Comments to the Authors (Required)):

The authors have adequately addressed my concerns and revised the manuscript accordingly.

Reviewer #2 (Comments to the Authors (Required)):

The authors have answered all my comments and concerns.

September 26, 2023

RE: Life Science Alliance Manuscript #LSA-2023-02129-TRR

Dr. Yoshihiko Miyata
Kyoto University
Graduate School of Biostudies, Department of Cell and Developmental Biology
Yoshida Hon-machi
Room 304, Research Building No.16, Kyoto University
Sakyo-ku, Kyoto 606-8501
Japan

Dear Dr. Miyata,

Thank you for submitting your Research Article entitled "Identification of FAM53C as a cytosolic-anchoring inhibitory binding protein of the kinase DYRK1A". It is a pleasure to let you know that your manuscript is now accepted for publication in Life Science Alliance. Congratulations on this interesting work.

DISTRIBUTION OF MATERIALS:

Again, congratulations on a very nice paper. I hope you found the review process to be constructive and are pleased with how the manuscript was handled editorially. We look forward to future exciting submissions from your lab.

Sincerely,
